# Changes in operation of postural networks in rabbits with postural functions recovered after lateral hemisection of the spinal cord

Pavel V. Zelenin, Vladimir F. Lyalka and Tatiana G. Deliagina

*Department of Neuroscience, Karolinska Institute, Stockholm, Sweden*

Handling Editors: Richard Carson & Jing-Ning Zhu

The peer review history is available in the Supporting Information section of this article (https://doi.org/10.1113/JP283458#support-information-section).

**Changes in operation of postural networks in rabbits with postural functions recovered after lateral hemisection of the spinal cord**

- Postural limb reflexes on the side of the lateral hemisection recover mainly due to changes at the brainstem-cerebellum-spinal levels, while forebrain is substantially involved in generation of postural limb reflexes on the side contralateral to hemisection.

- Changes underlying recovery of postural functions take place not only in postural networks that are severely impaired, but also in those that are almost unaffected by acute lateral hemisection.

**Abstract** Acute lateral hemisection of the spinal cord (LHS) severely impairs postural functions, which recover over time. Here, to reveal changes in the operation of postural networks underlying the recovery, male rabbits with recovered postural functions after LHS at T12 (R-rabbits) were used.

**Pavel Zelenin** is Senior Researcher at the Department of Neuroscience, Karolinska Institute (Stockholm, Sweden). His research is mainly devoted to neuronal networks underlying the basic motor functions, including locomotion and postural control. His studies address the normal functioning of these networks, changes in the networks when functions are impaired by different types of injuries, as well as the different possible approaches to restoration of the impaired functions.

After decerebration, we characterized the responses of individual spinal interneurons from L5 along with hindlimb EMG responses to stimulation causing postural limb reflexes (PLRs) that substantially contribute to postural corrections in intact animals. The data were compared with those obtained in our previous studies of rabbits with the intact spinal cord and rabbits after acute LHS. Although, in R-rabbits, the EMG responses to postural disturbances both ipsilateral and contralateral to the LHS (ipsi-LHS and co-LHS) were only slightly distorted, PLRs on the co-LHS side (unaffected by acute LHS) were distorted substantially and PLRs on the ipsi-LHS side (abolished by acute LHS) were close to control. Thus, in R-rabbits, plastic changes develop in postural networks both affected and unaffected by acute LHS. PLRs on the ipsi-LHS side recover mainly as a result of changes at brainstem–cerebellum–spinal levels, whereas the forebrain is substantially involved in the generation of PLRs on the co-LHS side. We found that, in areas of grey matter in which the activity of spinal neurons of the postural network was significantly decreased after acute LHS, it recovered to the control level, whereas, in areas unaffected by acute LHS, it was significantly changed. These changes underlie the recovery and distortion of PLRs on the ipsi-LHS and co-LHS sides, respectively.

(Received 19 August 2022; accepted after revision 30 November 2022; first published online 4 December 2022)

**Corresponding author** T. G. Deliagina: Department of Neuroscience, Karolinska Institute, Solnavägen 9, Stockholm 171 77, Sweden.      Email: tatiana.deliagina@ki.se

**Abstract figure legend** To reveal changes in postural networks underlying recovery of postural functions, in rabbits recovered after the lateral hemisection of the spinal cord, postural corrections caused by lateral tilt of the supporting platform were evaluated. Then, the rabbits were decerebrated, and the activity of individual spinal interneurons of postural networks (F-neurons and E-neurons) located below the lesion was recorded during postural limb reflexes. The results were compared with previously obtained control data from animals with an intact spinal cord, as well as with data obtained after acute lateral hemisection (not shown). We demonstrate that activity of neurons of postural networks does not return to the control level in recovered animals.

### Key points

- After lateral hemisection of the spinal cord (LHS), postural functions recover over time. The underlying changes in the operation of postural networks are unknown.
- We compared the responses of individual spinal neurons and hindlimb muscles to stimulation causing postural limb reflexes (PLRs) in recovered LHS-rabbits with those obtained in rabbits with the intact spinal cord and rabbits after acute LHS.
- We demonstrated that changes underlying the recovery of postural functions take place not only in postural networks that are severely impaired, but also in those that are almost unaffected by acute LHS.
- PLRs on the LHS side recover mainly as a result of changes at brainstem–cerebellum–spinal levels, whereas the forebrain is substantially involved in the generation of PLRs contralateral to the LHS.

## Introduction

Humans and most animals actively stabilize a particular body orientation in space because of the activity of the postural system. Spinal cord injury dramatically affects the operation of the postural system. Although complete spinalization abolishes postural functions and they practically do not recover with time (Barbeau et al. 2002; Chvatal et al. 2013; Lyalka et al. 2011; Macpherson & Fung, 1999; Macpherson et al. 1997; Rossignol et al., 1999, 2002), after some types of incomplete spinal cord injury, postural functions may recover. In particular, lateral hemisection of the spinal cord (LHS) at the low thoracic level causes a dramatic right-left asymmetry in the muscle tone and hindlimb reflexes, resulting in the inability to stand and walk. However, these functions mostly recover in a few weeks post lesion (Frigon & Rossignol, 2006; Helgren & Goldberger, 1993; Hultborn & Malmsten, 1983a, 1983b; Kuhtz-Buschbeck et al. 1996). We demonstrated that LHS at T12 in rabbits causes a loss of balance in the hindquarters in the acute state, but close-to-normal postural corrections in response to lateral tilts of the supporting surface recover in 2–3 weeks (Lyalka et al. 2005). Changes in the operation of postural neuronal networks underlying the recovery of postural functions after LHS are unknown.

In terrestrial quadrupeds, lateral stability is maintained as a result of the activity of the postural system, which is driven mainly by somatosensory information from limb mechanoreceptors (Beloozerova et al. 2003; Deliagina et al., 2000, 2006, 2012; Horak & Macpherson, 1996; Inglis & Macpherson, 1995; Stapley & Drew, 2009). It was demonstrated that the nervous mechanisms controlling the orientation of the anterior and posterior parts of the body can operate independently of each other and generate postural corrections mainly in response to somatosensory information from the same limb (Beloozerova et al. 2003; Deliagina et al. 2006).

Previously, we demonstrated that the basic postural networks underlying lateral stability reside at the brainstem-cerebellum-spinal cord levels (Musienko et al. 2008). In decerebrate rabbits, we characterized postural limb reflexes (PLRs), which cause a change in the activity of limb extensors. In intact animals, they substantially contribute to postural corrections aimed to ensure balance when standing (Deliagina et al., 2012, 2014; Musienko et al., 2008, 2010) and walking (Musienko et al. 2014). They are generated mainly in response to sensory inputs from the ipsilateral limb (Musienko et al. 2010). It was demonstrated that, although the spinal cord contains neuronal networks generating spinal PLRs, their efficacy is low and that supraspinal systems substantially contribute to the generation of PLRs (Deliagina et al. 2014; Musienko et al. 2010). Two groups of spinal interneurons (F-neurons and E-neurons) active in phase and in anti-phase, respectively, with their ipsilateral limb extensors during PLRs have been revealed (Hsu et al. 2012; Zelenin et al. 2015). It was suggested that at least some of the F- and E-neurons are pre-motor interneurons exciting and inhibiting extensor motoneurons, respectively (Zelenin et al. 2015).

Recently, we characterized the state of postural networks generating PLRs after acute LHS (Zelenin, Lyalka, Orlovsky et al. 2016), which represents a starting point for the development of plastic changes leading to the recovery of postural functions. We demonstrated that acute LHS causes the disappearance of PLRs on the ipsilateral to LHS (ipsi-LHS) side, a significant decrease in the activity of F- and E-neurons located in specific areas of the grey matter on both sides of the spinal cord, as well as specific changes in the efficacy of posture-related sensory inputs to them. By contrast, acute LHS hardly affects PLRs contralateral to LHS (co-LHS). It was suggested that this drastic right-left asymmetry in PLRs is the reason for lateral instability in rabbits with acute LHS (Lyalka et al. 2005; Zelenin, Lyalka, Orlovsky et al. 2016).

Here, aiming to reveal changes in postural networks underlying the recovery of postural functions, first, in LHS-rabbits with the spontaneously recovered ability to maintain balance when standing and walking (R-rabbits), postural corrections caused by the lateral tilt of the supporting platform were evaluated. Second, R-rabbits were decerebrated, and the responses of individual spinal interneurons from L5 along with hindlimb muscles EMG responses to stimulation causing PLRs in subjects with an intact spinal cord were recorded. The results were compared with control data from animals with the intact spinal cord (Zelenin et al. 2015), as well as with data obtained after acute LHS (Zelenin, Lyalka, Orlovsky et al. 2016). Our results suggest that changes underlying the recovery of postural functions take place not only in postural networks that are severely impaired, but also in those almost unaffected by acute LHS.

## Methods

### Ethical approval

Experiments were carried out on eight adult male New Zealand rabbits (*Oryctolagus cuniculus*) (weighing 2.5–3.0 kg). The animals were obtained from HB Lidköpings Kaninfarm (Lidköping, Sweden). All procedures were conducted in accordance with protocols (N53/15, 11769-2020) approved by the local ethical committee (Norra Djurförsöksetiska Nämden) in Stockholm and followed the European Community Council Directive (2010/63EU) and the guidelines of the National Institute of Health *Guide for the Care and Use of Laboratory Animals*. All authors understand the ethical principles under which *The Journal of Physiology* operates and this work complies with the animal ethics checklist.

### Surgical procedures

Each rabbit was subjected to two operations. The first surgery was performed under Hypnorm-midazolam anaesthesia [Hypnorm (fentanyl/fluanisone, VetaPharma Ltd, Leeds, UK) 0.3 mL kg$^{-1}$ I.M., midazolam 1 mg kg$^{-1}$ I.M.), using aseptic procedures. The level of anaesthesia was controlled by applying pressure to a paw (to detect limb withdrawal) and by examining the size and reactivity of pupils. During the first surgery, chronic implantation of bipolar EMG electrodes was performed bilaterally into musculus gastrocnemius lateralis (Gast, ankle extensor) and musculus vastus lateralis (Vast, knee extensor) using the method described previously (Lyalka et al., 2005, 2011). Then, an incision was made along the dorsal midline in the lower thoracic region and a laminectomy at the T11–T12 level was performed. The dura in the rostral part of the T12 segment was opened and a lateral hemisection on the left side of the spinal cord was achieved with spring scissors, microsurgery forceps and a small scalpel under a microscope. Afterward, the incision was closed in anatomic layers.

LHS-rabbits were not subjected to any training or specific behavioural testing during the recovery period.

In 49–93 days after LHS, in rabbits with spontaneously recovered ability to stand and to walk (R-rabbits), a capability to perform postural corrections was evaluated (see below, Postural test). Then, R-rabbits were taken to the second surgery for an acute experiment (see below, Acute experiment). The following terminal procedure was used. For induction of anaesthesia, the animal was injected with propofol (average dose, 10 mg kg$^{-1}$, administered I.V.). Afterward, it was continued on isoflurane (1.5–2.5%) delivered in $O_2$. The trachea was cannulated and laminectomy at L5 (exposing the spinal cord for recording of neurons) was performed. To insert the recording microelectrode, small holes ($\sim$1 mm$^2$) were made in the dura mater at L5. The rabbit was decerebrated at the precollicular–postmammillary level (Musienko et al. 2008) and then, the anaesthesia was discontinued. During experiments, the animals were spontaneously breathing with room air. The rectal temperature was maintained at 37–38°C with help of heat irradiation. The blood pressure was maintained at >80 mmHg. If needed, an injection of prednisolone (3 mg kg$^{-1}$, I.M.) was performed to stabilize the arterial pressure and to reduce the brain swelling after decerebration. Collection of data began 1.5–2 h after decerebration.

## Animal care

All rabbits were subjected to the same treatment after the first surgery and were kept in the same environmental conditions. Each animal was individually caged and were provided with food (dry rabbit food, hay and carrots) and water *ad libitum*. The bottom of the cage was covered by soft absorbing tissue. The animals were monitored attentively. Every 12 h for 48 h after surgery, the rabbits received an analgesic (Temgesic, RB Pharmaceuticals Limited, Slough, UK; 0.01 mg kg$^{-1}$ s.c.). In addition, Rimadyl (Orion Pharma Animal Health, Sollentuna, Sweden; 4 mg kg$^{-1}$ s.c.) was administered before surgery and 2 days after surgery to reduce inflammation. The first 2 days after LHS, the animals received 25 mL of Ringer solution (s.c.) twice a day. Urination was monitored and the hindquarters were inspected and cleaned if necessary. R-rabbits usually regained the ability to stand and to walk within 3–7 days after surgery. R-rabbits had an asymmetrical body configuration similar to that described previously (Lyalka et al. 2005). After LHS, a twisting of the caudal part of the trunk toward the co-LHS side gradually developed and the hindlimb on the co-LHS side was turned inward and occurred at the abnormal medial position.

## Experimental design

**Postural test.** To evaluate the efficacy of postural corrections, R-rabbits were subjected to a postural test on a tilting platform (Beloozerova et al. 2003; Deliagina et al. 2000; Lyalka et al. 2011) before the acute experiment. No special training of the rabbits was required prior to testing. For testing, a rabbit was positioned on the two platforms (P1 and P2 in Fig. 1*A*), so that P1 supported the forelimbs and P2 supported the hindlimbs. The sagittal plane of the rabbit was aligned to the axis of the platforms rotation (Fig. 1*A* and *B*). The surface of the platforms was covered with sandpaper to prevent sliding of the animal during tilts. Previously, we demonstrated that relatively independent neural mechanisms generate postural corrections stabilizing the dorsal side up orientation of the anterior and posterior parts of the trunk (Beloozerova et al. 2003; Deliagina et al. 2006). To evaluate postural corrections in the hindquarters (below the LHS), the P2 platform was tilted in the frontal (transverse) plane of the animal (angle $\alpha$) (Fig. 1*A* and *C*) with the amplitude ±20° (Fig. 1*H*), whereas the P1 platform was kept horizontal. This allowed postural disturbances to be applied primarily to the hindquarters. A trapezoidal time profile of the platform tilts was used with a period of $\sim$6 s (Fig. 1*H*), transition between extreme positions lasted for $\sim$1 s and each extreme position was maintained for $\sim$2 s. Previously, we demonstrated that lateral displacement of the trunk in relation to the tilting platform characterizes the efficacy of stabilization of the dorsal side-up trunk orientation (Beloozerova et al. 2003). To evaluate efficacy of postural corrections below LHS, lateral displacements of the caudal part of the trunk in relation to the P2 platform (the corrective trunk movements) were monitored by a sensor (S in Fig. 1*A*–*C*). The sensor consisted of a high-resolution variable resistor the axis of which was rotated by means of a long lever; the latter was touching the lateral aspect of the body at the half body height. To detect motion of the body in relation to the tilting platform in both directions, a lever was kept pressed to the body with a soft spring. During the postural test, EMGs from selected hindlimb muscles were recorded along with the tilt angle of the platform and corrective trunk movements. All eight R-rabbits were subjected to this test. In total, 902 individual EMG responses were recorded and used for the analysis.

**Acute experiment.** After postural test, R-rabbits were taken to an acute experiment. The experimental design for the acute experiments was similar to that described previouslu (Hsu et al. 2012; Zelenin et al. 2015; Zelenin, Lyalka, Orlovsky et al. 2016). The head and vertebral column of the decerebrate rabbit were rigidly fixed; the forelimbs were suspended in a hammock (Fig. 1*D*). To evoke PLRs, the hindlimbs of the rabbit were positioned on a horizontal platform, with limb configuration and the inter-feet distance similar to that observed in freely standing rabbits (Beloozerova et al. 2003). The platform

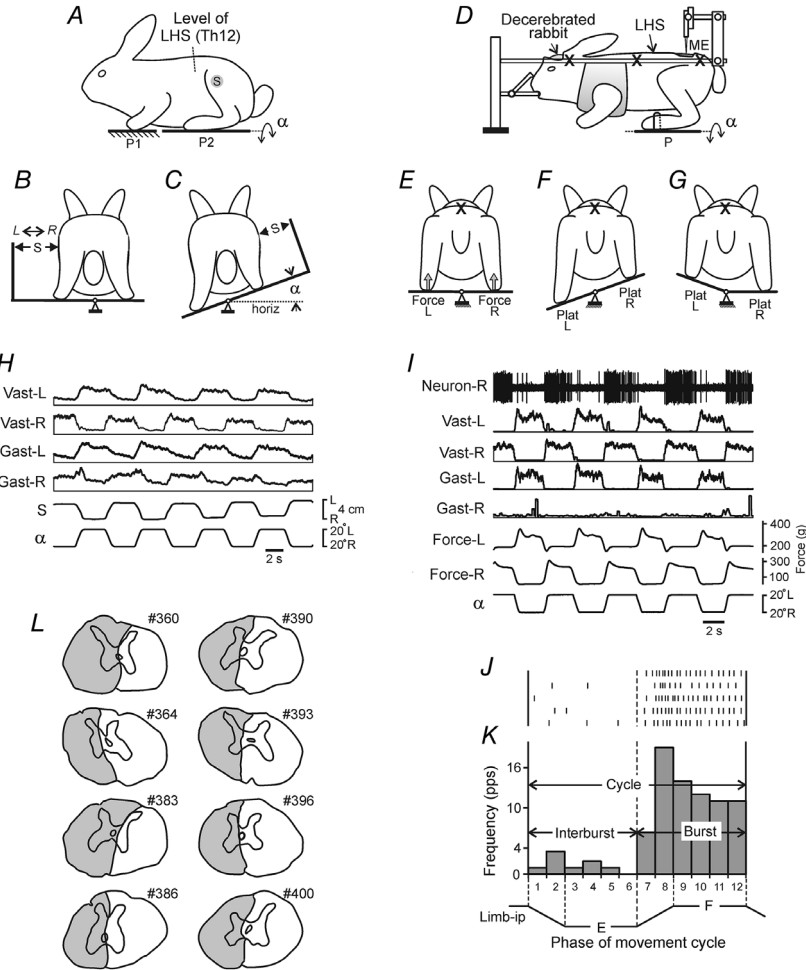

**Figure 1. Experimental design**

*A–C*, testing postural reactions to tilts. The animal was standing on two platforms: one under the forelimbs and another one under the hindlimbs (*P1* and *P2*, respectively, in *A*). Platform under the hindlimbs (*P2*) could be tilted in the transverse plane (α is the platform tilt angle) (*C*). The sagittal plane of the animal was aligned with the axis of platform rotation. A mechanical sensor (*S*) positioned at the half-height of the body (*A*), measured lateral displacements of the caudal part of the trunk in relation to the platform *P2* (*B* and *C*). *D–G*, experimental design for acute experiments. The decerebrate rabbit was fixed in a rigid frame (points of fixation are indicated by 'X'). To evoke postural limb reflexes (PLRs), the hind limbs were positioned on a platform (*P* in *D*), which was periodically tilted in the transversal plane either as a whole (*F*) or its left (*Plat L*) and right (*Plat R*) parts were tilted separately (*G*). The contact forces under the left and right limbs were measured by force sensors (*Force L* and *Force R* in *E*). Activity of spinal neurons from L5 was recorded by means of a microelectrode (*ME* in *D*). *H* and *I*, postural responses to tilts in R-rabbit #386 recorded before decerebration (*H*) and in the acute experiment after decerebration (*I*). In (*I*), activity of a neuron located on the right (co-LHS) side of the spinal cord (*Neuron-R*) was recorded simultaneously with PLRs. The neuron was activated with flexion of the ipsilateral limb (F-neuron) caused by the platform tilt to the left. In *H* and *I*, the EMGs of the left and right musculus gastrocnemius lateralis (Gast-L and Gast-R, respectively) and musculus vastus lateralis (Vast-L and Vast-R, respectively) were recorded. Note that the phases of the correct EMG response in the tilt cycle in the freely standing (*H*) and in the fixed decerebrate R-rabbit #386 (*I*) are opposite (for explanation, see text). In the freely standing R-rabbit #386, Gast and Vast on both the ipsi-LHS (left) side and co-LHS (right) side had correct responses to tilts (activation with the ipsilateral tilt). In the fixed decerebrate R-rabbit #386 (*I*), Vast-L and Gast-L on the ipsi-LHS side and Vast-R on the co-LHS side had correct responses (activation with the contralateral tilt), whereas Gast-R on the co-LHS side did not respond to tilts. *J* and *K*, a raster of responses of the neuron shown in (*I*) in five sequential movement cycles of the ipsilateral limb and a histogram of its spike activity in different phases (1–12) of the cycle of movement (E, extension; F, flexion) of the ipsilateral limb (Limb-ip). The neuron was activated with flexion of the ipsilateral limb (F-neuron). The halves of the cycle with higher (F, bins 7–12) and lower (E, bins 1–6) neuronal activity were designated as 'burst' and 'interburst' periods, respectively. *L*, extent of the spinal cord damage in individual R-rabbits. The total extent of the lesion (shaded area) is projected on a spinal cord section taken more rostrally after inspecting several consecutive sections.

as a whole, or its right or left parts separately, could be tilted periodically by rotation around the medial axis with the amplitude ±20° (Fig. 1*F* and *G*). Because the vertebral column and pelvis were fixed, tilts of the whole platform led to flexion/extension movements of the hip, knee and ankle joints and close-to-vertical displacements of the distal point of the limb. A trapezoidal time profile of the platform tilts was used causing trapezoidal limb displacement with a period of ∼6 s (Fig. 1*I*), transition between extreme positions lasted for ∼1 s, and each extreme position was maintained for ∼2 s. The tilt angle of each platform was monitored with a mechanical sensor. The contact forces under the limbs were measured by means of force sensors (Force in Fig. 1*I*). In subjects with an intact spinal cord, the tilt-related somatosensory stimulation (caused by loading and flexion of the limb on the platform side moving up and unloading and extension of the limb on the platform side moving down) evoked PLRs, which included activation of extensors in the flexing limb and an increase in its contact force, as well as inactivation of extensors in the extending limb and a decrease in its contact force (Hsu et al. 2012; Musienko et al. 2010). All eight R-rabbits were subjected to these postural tests. EMG responses to 334 tilts of the whole platform, 332 tilts of a platform only under a limb on the ipsi-LHS side and 326 tilts of a platform only under a limb on the co-LHS side were recorded and used for the analysis. In the acute experiments, the activity of individual spinal neurons during PLRs was recorded (see below).

### Recording and data analysis

EMG activity and signals from sensors were digitized (at 5 and 1 kHz, respectively) and saved to disk using the data acquisition and analysis system (Power1401/Spike2; Cambridge Electronic Design, Cambridge, UK). The EMG signals were rectified and smoothed (time constant, 100 ms).

Neurons were recorded extracellularly from the spinal segment L5 by means of commercially available varnish-insulated tungsten electrodes (shaft diameter 75 μm; impedance 4 to 7 MΩ; FHC, Bowdoin, ME, USA). We tended to explore systematically the whole cross-section of the left and right grey matter except for the areas of motor nuclei (Portal et al. 1991). The lateral and vertical co-ordinates of each neuron were marked on the map of the spinal cord cross-section (Hsu et al. 2012; Shek et al. 1986; Zelenin et al. 2015). Unfortunately, it was impossible to explore systematically the entire grey matter in individual decerebrate rabbits. Accordingly, we pooled all neurons recorded on the ipsi-LHS side and all neurons recorded on the co-LHS side from eight rabbits and did not perform any cross-subject comparison.

Neuronal data were digitized at 30 kHz and saved to disk simultaneously with the EMG activity (digitized at 5 kHz) and behavioural (position and force) markers (digitized at 1 kHz) using the data acquisition and analysis system (Power1401/Spike2; Cambridge Electronic Design). This system also allowed the waveform analysis to discriminate and identify the spikes of a single neuron using the waveform-matching algorithm. Only neurons with a stable spike shape were used for analysis. All eight R-rabbits were used in these acute experiments. In total, the responses to tilts of the whole platform were recorded and analysed in 185 neurons on the ipsi-LHS side and 194 neurons on the co-LHS side. Fuerthermore, responses to tilts of only ipsi- and only contralateral platforms were recorded and analysed in 110 modulated neurons on the ipsi-LHS side and 102 neurons on the co-LHS side (Table 1).

When processing the recorded data, we considered the activity of each neuron in the movement cycle of the ipsilateral limb because the phase of modulation in the majority of neurons was determined by the tilt-related sensory input from the ipsilateral limb (Hsu et al. 2012; Zelenin et al. 2015). For each individual neuron, a raster of activity in sequential movement cycles was obtained. The raster for one of the neurons is shown in Fig. 1*J*. The cycle was divided into 12 bins; the onset of extension of the ipsilateral limb was taken as the cycle onset. Bins 1 to 2 corresponded to extension of the limb; bins 3−6 corresponded to maintenance of the extended position; bins 7−8 corresponded to flexion of the limb; and bins 9−12 corresponded to maintenance of the flexed position. The firing frequency in each bin was calculated and averaged over the identical bins in all cycles at a given condition, and the phase histogram was generated (Fig. 1*K*). The mean frequency during extension of ipsilateral limb (bins 1−6) and that during flexion (bins 7−12) were compared. The larger and the smaller values were termed the burst frequency ($F_{BURST}$) and the interburst frequency ($F_{INTER}$), respectively. A neuron was considered to be modulated by tilts if the difference between the mean burst frequency and mean interburst frequency was statistically significant (two-tailed Welch's *t* test, $P < 0.05$). In addition, the mean frequency (average value over bins 1−12) and the depth of modulation ($M = F_{BURST} - F_{INTER}$) were calculated. To characterize quantitatively the activity of F- and E-neurons, each of the main parameters of activity (the mean frequency, the depth of modulation, the burst frequency and the interburst frequency) was averaged separately over F- or E-neurons in each of three zones of the grey matter (zone 1 corresponded to the dorsal part of the dorsal horn, zone 3 corresponded to the ventral horn and zone 2 corresponded to the intermediate area of the grey matter) (Fig. 3*A* and *B*).

**Table 1. The number of neurons subjected to different tests**

| Type of rabbits | Side relative to LHS | Type of neuron | Whole platform tilts | Ip/Co platform tilts | Receptive field |
|---|---|---|---|---|---|
| | | | Test | | |
| Control | Not | F | 249 | 175 | 286 |
| *N* = 20 | applicable | E | 186 | 132 | |
| | | N/M | 64 | – | – |
| | | *Total* | *499* | *307* | *286* |
| Acute LHS | Ipsi-LHS | F | 75 | 70 | 140 |
| *N* = 7 | side | E | 80 | 74 | |
| | | N/M | 63 | – | – |
| | | *Total* | *218* | *144* | *140* |
| | Co-LHS | F | 122 | 112 | 180 |
| | side | E | 87 | 79 | |
| | | N/M | 63 | – | – |
| | | *Total* | *272* | *191* | *180* |
| R-rabbits | Ipsi-LHS | F | 71 | 54 | 73 |
| *N* = 8 | side | E | 56 | 46 | |
| | | N/M | 58 | – | – |
| | | *Total* | *185* | *110* | *73* |
| | Co-LHS | F | 77 | 61 | 74 |
| | side | E | 53 | 41 | |
| | | N/M | 64 | – | – |
| | | *Total* | *194* | *102* | *74* |

The number of F-, E- and non-modulated neurons (F, E and N/M, respectively) in which responses to the whole platform tilts were recorded, and the responses to tilts of the ipsilateral platform only and to tilts of the contralateral platform only (Ip/Co platform tilts) were recorded, as well as the total number of modulated neurons subjected to receptive fields testing in each of three groups of animals (Control, animals with acute LHS, and R-rabbits). In animals with acute LHS and R-rabbits, neurons were counted separately on the ipsi-LHS and co-LHS sides. *N*, number of animals. The data related to control and to acute LHS were taken from the database of our previously published studies (Zelenin et al. 2015 and Zelenin, Lyalka, Orlovsky et al. 2016, respectively).

To reveal changes in the activity of local neuronal populations in different areas of the grey matter in R-rabbits, 'heatmaps' for the mean frequency and the depth of modulation in the control, after acute LHS, as well as in R-rabbits were generated. To calculate a value of the parameter in a point with co-ordinates (*x*,*y*) on the heatmap, values of the parameter for the neurons recorded in close proximity to this point were used. Depending on the distance *d* from recording point to the point (*x*,*y*), these values were weighted [Gaussian weighting $w(d) = exp(-d^2/D^2)$ with the spatial constant $D = 0.4$ mm]. To reveal changes in the parameter, subtraction of the heatmap for the control from the heatmap for rabbits after acute LHS and from the heatmap for R-rabbits was performed (Fig. 6). Areas of significant local changes (*t* test, $P < 0.05$) were delimited by solid lines (Fig. 6).

Peripheral receptive fields were examined in a portion of the recorded neurons. Stimuli included light brushing of hairs, light tapping of hair skin, firm tapping on and palpation of muscle bellies (flexors and extensors of ankle, knee and hip, and abductors and adductors of hip) and tendons, light pinching the skin with fingers, and, in a few cases, manual movements of joints. If a cutaneous input was observed, then the responses from the underlying muscles were not taken into account. Stimuli that could potentially activate nociceptors (i.e. pinching or poking with sharp instruments) were not used. In total, peripheral receptive fields of 73 modulated neurons on the ipsi-LHS side and 74 modulated neurons on the co-LHS side recorded in all eight R-rabbits were examined (Table 1).

The data for F-, E- and non-modulated neurons, as well as the EMG responses, obtained in experiments on R-rabbits were compared with the control data and with the data obtained after acute LHS taken from the database of our previously published studies (Zelenin et al. 2015 and Zelenin, Lyalka, Orlovsky et al. 2016, respectively). Corresponding numbers of animals and F-, E- and N/M neurons used for comparison are presented in Table 1. The experimental subjects, as well as all methods used in the control study (except for LHS, Zelenin et al. 2015) and in the study devoted to acute LHS (Zelenin, Lyalka, Orlovsky et al. 2016), were similar to those used in the present study.

## Statistical analysis

All quantitative data in this study are presented as the mean $\pm$ SD. Welch's $t$ test (two-tailed) was used to characterize the statistical significance when comparing different means. To evaluate the statistical significance of difference in relative numbers of different types of EMG responses to tilts and in relative numbers of neurons in different functional groups, we used Pearson's chi-squared test. To evaluate the statistical significance for linear regression, we used ANOVA. For all tests, the significance level was set at $P = 0.050$.

## Histological procedures and evaluation of the extent of spinal lesions

To verify positions of recording sites on the cross-section of the spinal cord, at the end of each experiment, we made reference electrolytic lesions in L5. Then, rabbits were deeply anesthetized with isoflurane (2.5%), and perfused transcardially with isotonic saline followed by 4% paraformaldehyde solution that resulted in death. A piece of the spinal cord containing these lesions, as well as a piece with the site of LHS, was fixed with 10% formalin solution, frozen and cut to sections of 30 $\mu$m thickness in the region of recording and in the region of LHS. The sections were stained with cresyl violet. Locations of recording sites were verified in relation to the reference lesions. The position and extent of LHS were verified by observation of a series of magnified digital images of sections (Fig. 1*L*).

Figure 1*L* shows the reconstructed lesion sites from eight R-rabbits. In all rabbits, the lesion occupied about half of the spinal cord cross-section. The medial edge of the lesion was close to the sagittal plane, with some 'under-cut' in five rabbits (#364, 390, 393, 396 and 400) and 'overcut' in two rabbits (#360 and 383).

## Results

### Postural limb reflexes in R-rabbits

First, to evaluate the efficacy of postural corrections in the hindquarters and EMG responses to tilts, R-rabbits were subjected to a postural test (for details, see Methods). In general, postural corrections in R-rabbits did not differ from those observed in intact animals (Beloozerova et al. 2003; Musienko et al. 2008). The tilt of the P2 platform caused an extension of the hindlimb on the side moving downward and a flexion of the limb on the opposite side (as shown schematically in Fig. 1*C*). These hindlimb movements resulted in the displacement of the caudal part of the trunk in the direction opposite to the tilt (postural correction, S). Figure 1*H* shows a representative example of postural corrections and EMG responses in an R-rabbit

(#386). Body displacements (S) were in antiphase to tilts ($\alpha$), indicating the presence of postural corrections. EMG responses of Vast and Gast on both the ipsi-LHS and co-LHS sides were also correctly phased (activation with the ipsilateral tilt).

To quantitatively evaluate the efficacy of postural corrections in R-rabbits, we calculated the gain of postural corrections defined as $G = S_{PP}/\alpha_{PP}$ (cm/°), where $S_{PP}$ is the peak-to-peak value of postural corrections and $\alpha_{PP}$ is the peak-to-peak value of the platform tilt. As seen in Fig. 2*B*, the average gain of postural corrections in R-rabbits was similar to that in intact rabbits (control) (Welch's $t$ test, $P = 0.526$). Despite similarities in the gain, the phase of EMG response in the tilt cycle was slightly less consistent than in intact rabbits (Fig. 2*A*). Although, in intact rabbits (control), Vast and Gast had a correct response (activation with the ipsilateral tilt) in 100% of tilt cycles, in R-rabbits, an incorrect response (activation with the contralateral tilt), a correct/incorrect response (activation with both the ipsilateral and contralateral tilts) and an absence of any response were observed in the hindlimb on the ipsi-LHS side in 23% of cycles (different from control, Welch's $t$ test, $P = 9 \times 10^{-4}$) and on the co-LHS side in $\sim$21% of cycles (different from control, Welch's $t$ test, $P = 0.033$). However, in R-rabbits in the overwhelming majority of tilt cycles, EMG responses were correct in both hindlimbs (Fig. 2*A*).

Second, we evaluated EMG responses in Vast and Gast to whole platform tilts in decerebrated R-rabbits. It should be noted that, in the decerebrate rabbit with an intact spinal cord and a fixed spine, the correct phase of the EMG response in the tilt cycle is opposite to that observed in the intact, freely standing rabbit (Beloozerova et al. 2003; Musienko et al., 2008, 2010). In both the freely standing rabbit and the decerebrate preparation, the activation and inactivation of limb extensors are caused mainly by loading and unloading the limb, respectively. However, in the freely standing rabbit, the tilt causes the loading of the limb moving down, as well as simultaneous unloading of the opposite limb, whereas, in the decerebrate rabbit, because of the fixed spine, the tilt causes the loading and flexion of the limb moving up, as well as unloading and extension of the opposite limb. Thus, in the freely standing rabbit, the correct EMG response is activation with the ipsilateral tilt, whereas, in the decerebrate rabbit, the correct EMG response is activation with the contralateral tilt.

Figure 1*I* shows EMG responses to tilts in R-rabbit #386 after decerebration. On the ipsi-LHS (left) side, both Gast-L and Vast-L were activated with contralateral (right) tilts and thus their responses were correct. However, on the co-LHS (right) side, only Vast-R responded correctly, whereas Gast-R did not respond to the tilts. In Figure 2*C*, the proportions of different types of muscle responses observed in the control, after acute LHS and

in R-rabbits are compared. After acute LHS, the correct muscle responses (activation with the contralateral tilt) were observed only in 25% of cycles *vs.* 100% of cycles in the control. In R-rabbits, they were mostly restored (the correct responses were observed in 74% of tilt cycles,

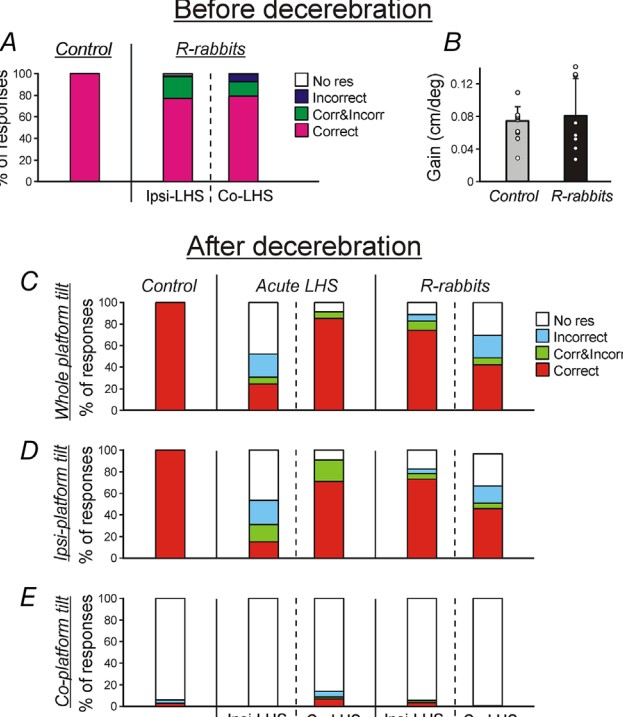

**Figure 2. Motor responses to tilts in the control and LHS rabbits, as well as in the corresponding decerebrate preparations**

*A*, proportion of different types of EMG responses to the whole platform tilts in Vast and Gast recorded in the rabbits with an intact spinal cord (Control, $N = 9$, $n = 264$) and in R-rabbits ($N = 8$, $n = 902$) in the limb on the ipsi-LHS side (*Ipsi-LHS*) and in the limb on the co-LHS side (*Co-LHS*). *Correct*, activation with ipsilateral tilt; *Incorrect*, activation with contralateral tilt; *Corr&Incorr*, activation with both ipsi- and contra-tilts; *No res*, no EMG response to tilt. *B*, the gain of postural corrections in the control and R-rabbits ($N = 9$ and $N = 8$, respectively). *C*, proportion of different types of EMG responses to the whole platform tilts in Vast and Gast in decerebrate rabbits before (*Control*) and after acute LHS, as well as in R-rabbits in the limb on the ipsi-LHS (*Ipsi-LHS*) and co-LHS (*Co-LHS*) side. (*Flex*, activation with ipsi-limb flexion; *Ext*, activation with limb extension; *Fl&Ex*, activation with both movements; *No res*, no EMG response). Number of animals: $N = 7$ (before LHS and after acute LHS); $N = 8$ (R-rabbits). *D* and *E*, proportion of different types of EMG responses to separate tilts of the ipsilateral (*D*) and contralateral (*E*) platform (i.e. platform under the limb and under the contralateral limb, respectively) before (*Control*) and after acute LHS, as well as in R-rabbits. In (*D*), number of animals for the control: $N = 7$; for acute LHS: $N = 7$; for R-rabbits: $N = 8$. In (*E*), number of animals for the control: $N = 7$; for acute LHS: $N = 7$; for R-rabbits: $N = 8$. The mean ± SD values for a population were calculated across mean values from individual animals (the mean for an individual animal was calculated over 10–20 cycles). [Colour figure can be viewed at wileyonlinelibrary.com]

which was significantly higher than it was after acute LHS; Welch's *t* test, $P = 0.022$). Surprisingly, on the co-LHS side in R-rabbits, correct EMG responses were observed only in 42% of tilt cycles, which was significantly lower than it was after acute LHS (Welch's *t* test, $P = 0.001$), whereas, in the majority of cycles, the muscles did not respond to tilts, had an incorrect response (activation with the ipsilateral tilt) or correct/incorrect responses (responding to both the ipsilateral and contralateral tilts). These results suggest that in the process of functional recovery after LHS, plastic changes occurred in networks generating PLRs on the ipsi-LHS side and in networks generating PLRs on the co-LHS side.

To estimate the role of tilt-related sensory information from different hindlimbs in the generation of muscle responses in R-rabbits, we recorded EMG responses to the separate tilts of the left and right platforms. As in the control and after acute LHS, in R-rabbits, the hindlimbs on both the ipsi-LHS and co-LHS sides generated responses to the tilts of the ipsilateral platform (Fig. 2*D*) similar to those observed in response to the whole platform tilts (Fig. 2*C*) (Welch's *t* test, $P = 0.957$ and $P = 0.852$, respectively). By contrast to this, responses to the tilts of the contralateral platform (Fig. 2*E*) were practically absent (in sharp contrast to the tilts of the whole platform; Welch's *t* test, $P = 0.0013$ for the ipsi-LHS side, $P = 0.008$ for the co-LHS side). Thus, incorrectly phased responses in R-rabbits on the co-LHS side are caused by distortion in the processing of tilt-related sensory information from the ipsilateral limb.

## Proportion of different types of neurons recorded on the ipsi-LHS and co-LHS sides in R-rabbits

In eight decerebrate R-rabbits, 379 neurons were recorded during periodical tilts of the whole platform, causing passive flexion/extension limb movements. On the ipsi-LHS (ipsilateral to LHS) and co-LHS (contralateral to LHS) sides, 185 and 194 neurons were recorded, respectively (Table 1). They were considered putative interneurons becaus the majority of these neurons were recorded outside the motor nuclei area (delineated by dotted lines in Fig. 3*A* and *B*).

To reveal the changes in spinal postural networks in R-rabbits, the data obtained in the present study were compared with data from our previous studies obtained in decerebrate rabbits with the intact spinal cord (control data; $N = 20$, $n = 499$; Zelenin et al. 2015) and in decerebrate rabbits after acute LHS ($N = 7$, $n = 218$ and $n = 272$ on the ipsi-LHS and co-LHS sides, respectively; Zelenin, Lyalka, Orlovsky et al. 2016) (Table 1).

Figure 3*A* and *B* shows the location of individual F- and E-neurons (Figure 3*A*) as well as non-modulated neurons (Figure 3*B*) on the cross-section of the ipsi-LHS and

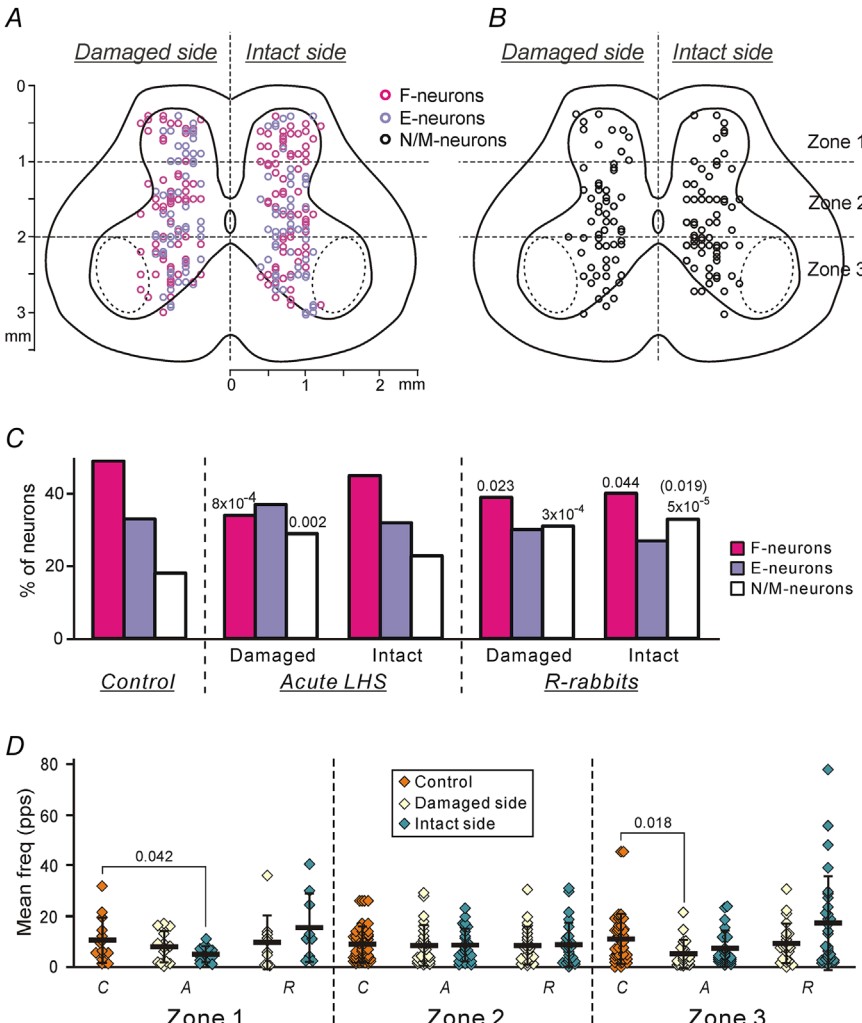

**Figure 3. Neurons recorded in R-rabbits**
*A* and *B*, position of F- and E-neurons (*A*), as well as non-modulated neurons (*B*), on the cross-section of the spinal cord recorded on the ipsi-LHS (ipsilateral to LHS) and on the co-LHS (contralateral to LHS) side in R-rabbits. *C*, relative number of F-, E- and non-modulated neurons recorded in control, as well as on the ipsi-LHS and on the co-LHS side of the spinal cord, after acute LHS and in R-rabbits. In (*A*) to (*C*), the number of animals and the number of F-, E- and non-modulated neurons recorded on the ipsi-LHS and co-LHS side in R-rabbits: $N = 8$, $n = 71$, 56 and 58, and $n = 77$, 53 and 64, respectively. In (*C*), the number of animals and the number of F-, E- and non-modulated neurons in the control: $N = 15$, $n = 175$, 121 and 64, respectively; the number of animals and the number of F-, E- and non-modulated neurons recorded on the ipsi-LHS and co-LHS side after acute LHS: $N = 7$, $n = 75$, 80 and 63, and $n = 122$, 87 and 63, respectively. Note that, for the control, the data are presented that were obtained in rabbits in which not only modulated neurons, but also non-modulated neurons were systematically recorded. *D*, mean ± SD values of the mean frequency of non-modulated neurons (indicated by black lines across corresponding clouds of diamonds) recorded in the control (*C*), after acute LHS (*A*) and in R-rabbits (*R*) on the ipsi-LHS and on the co-LHS side are shown for the subpopulations located in different zones (1–3 shown in *A* and *B*) of the grey matter. The number of animals and the number of non-modulated neurons recorded in zones 1–3 in the control: $N = 15$, $n = 14$, 27 and 23, respectively. The number of animals and the numbers of non-modulated neurons recorded in zones 1–3 on the ipsi-LHS and co-LHS side after acute LHS: $N = 7$, $n = 16$, 27 and 20, and $n = 13$, 26 and 24, respectively; in R-rabbits: $N = 8$, $n = 10$, 24 and 24, and $n = 9$, 25 and 30, respectively. The *P* values are indicated when there is a statistically significant difference as compared to the control. The *P* value in parenthesis is a comparison of values in R-rabbit and in rabbits after acute LHS. [Colour figure can be viewed at wileyonlinelibrary.com]

co-LHS sides of the spinal cord. As in the control (Zelenin et al. 2015; Hsu et al. 2012) and after acute LHS (Zelenin, Lyalka, Orlovsky et al. 2016), in R-rabbits, F-neurons (active in phase with ipsilateral limb extensors during PLRs), E-neurons (active in antiphase with ipsilateral limb extensors during PLRs) and non-modulated neurons were found on both sides of the spinal cord and were intermixed (Fig. 3*A* and *B*).

On both the ipsi-LHS and co-LHS sides of the spinal cord in R-rabbits, the percentage of F-neurons was significantly lower (chi-squared test, $P = 0.023$ and 0.044, respectively), the percentage of non-modulated neurons was significantly higher (chi-squared test, $P = 3 \times 10^{-4}$ and $5 \times 10^{-5}$, respectively), whereas the percentage of E-neurons was similar to that in the control (chi-squared test, $P = 0.430$ and 0.128) (Fig. 3*C*). The relative numbers of F-, E- and non-modulated neurons on the ipsi-LHS side in R-rabbits were similar to those observed on the ipsi-LHS side after acute LHS (chi-squared test, $P = 0.408$, 0.174, and 0.593, respectively) (Fig. 3*C*). On the co-LHS side, proportions of F- and E-neurons in R-rabbits also did not differ significantly from those observed on the co-LHS side after acute LHS (chi-squared test, $P = 0.267$ and 0.279, respectively), whereas the proportion of non-modulated neurons was significantly higher than that after acute LHS (chi-squared test, $P = 0.019$) (Fig. 3*C*).

### Activity of neurons on the ipsi-LHS and co-LHS sides in R-rabbits

The activity of F- and E-neurons was characterized by four parameters: the mean frequency of a neuron, burst frequency, interburst frequency and depth of modulation (see Methods). To quantitatively characterize the activity of F- and E-neurons, each parameter was averaged separately over F- or E-neurons in each of the three zones (zone 1 corresponds to the dorsal part of the dorsal horn, zone 3 corresponds to the ventral horn and zone 2 corresponds to the intermediate area of the grey matter) (Fig. 3*A* and *B*).

**F-neurons.** In R-rabbits, most of the main activity parameters in F-neurons on the ipsi-LHS side of the spinal cord (Fig. 4), which exhibited a significant decrease after acute LHS (i.e. the mean frequency, the depth of modulation, the burst and interburst frequency in zone 3, and the depth of modulation in zone 2) were similar to the control (Welch's *t* test, respectively, $P = 0.125$, 0.168, 0.083 and 0.275 for the parameters in zone 3; $P = 0.675$ for the depth of modulation in zone 2) (Fig. 4), except for the interburst frequency in R-rabbits in zone 1, which remained significantly decreased (Fig. 4*D*). In addition, in R-rabbits, a significant decrease in the mean frequency (Fig. 4*A*) and burst frequency (Fig. 4*C*) on the

ipsi-LHS side in zone 1 was observed. On the co-LHS side of the spinal cord in R-rabbits, as after acute LHS, all parameters of F-neuron activity did not differ significantly from control in any of the three zones (Welch's *t* test, respectively, in zones 1, 2 and 3: $P = 0.874$, 0.152 and 0.172 for the mean frequency; $P = 0.220$, 0.348 and 0.051 for the depth of modulation; $P = 0.575$, 0.184 and 0.079 for the burst frequency; $P = 0.418$, 0.271 and 0.778 for the interburst frequency) (Fig. 4).

**E-neurons.** In R-rabbits on the ipsi-LHS side, the values of most activity parameters of E-neurons located in zone 1, which were significantly decreased after acute LHS, did not differ from control (Welch's *t* test, $P = 0.953$ for the mean frequency; $P = 0.450$ for the depth of modulation; $P = 0.718$ for the burst frequency) (Fig. 5*A*–*C*). However, on the co-LHS side in zone 1, the mean frequency, the depth of modulation and the burst frequency were significantly lower than in the control and similar to those observed after acute LHS (Welch's *t* test, $P = 0.829$, 0.818 and 0.813, respectively) (Fig. 5*A*–*C*). In addition, in R-rabbits, the depth of modulation was significantly increased in E-neurons located on the co-LHS side in zone 3 (Fig. 5*B*).

**Non-modulated neurons.** In R-rabbits, the mean frequency of non-modulated neurons did not significantly differ from control on the ipsi-LHS, as well as on the co-LHS side of the spinal cord in each of the three zones (Welch's *t* test, respectively, in zones 1, 2 and 3: $P = 0.847$, 0.817 and 0.488 on the ipsi-LHS side and $P = 0.373$, 0.912 and 0.122 on the co-LHS side), although it was significantly decreased after acute LHS on the co-LHS side in zone 1 and on the ipsi-LHS side in zone 3 (Fig. 3*D*).

**Changes in the activity of local populations of F- and E-neurons.** To more precisely delineate the areas of grey matter containing populations of neurons exhibiting a significant change in activity compared to the control, in R-rabbits and rabbits after acute LHS, a subtraction of the control heatmaps (see Methods) for the mean frequency and the depth of modulation from the corresponding heatmaps of R-rabbits and rabbits after acute LHS was performed (Fig. 6). The local populations with a significant change ($P < 0.05$) in the parameter value are delineated by a continuous line.

On the ipsi-LHS side in R-rabbits, changes in the activity of F- and E-neurons were observed mostly in those areas of grey matter that were significantly affected by the acute LHS. Thus, the mean frequency and the depth of modulation of E-neurons located in areas in which these parameters of E-neurons were significantly reduced after acute LHS mostly returned to the control level (except for two small areas located in zones and zone

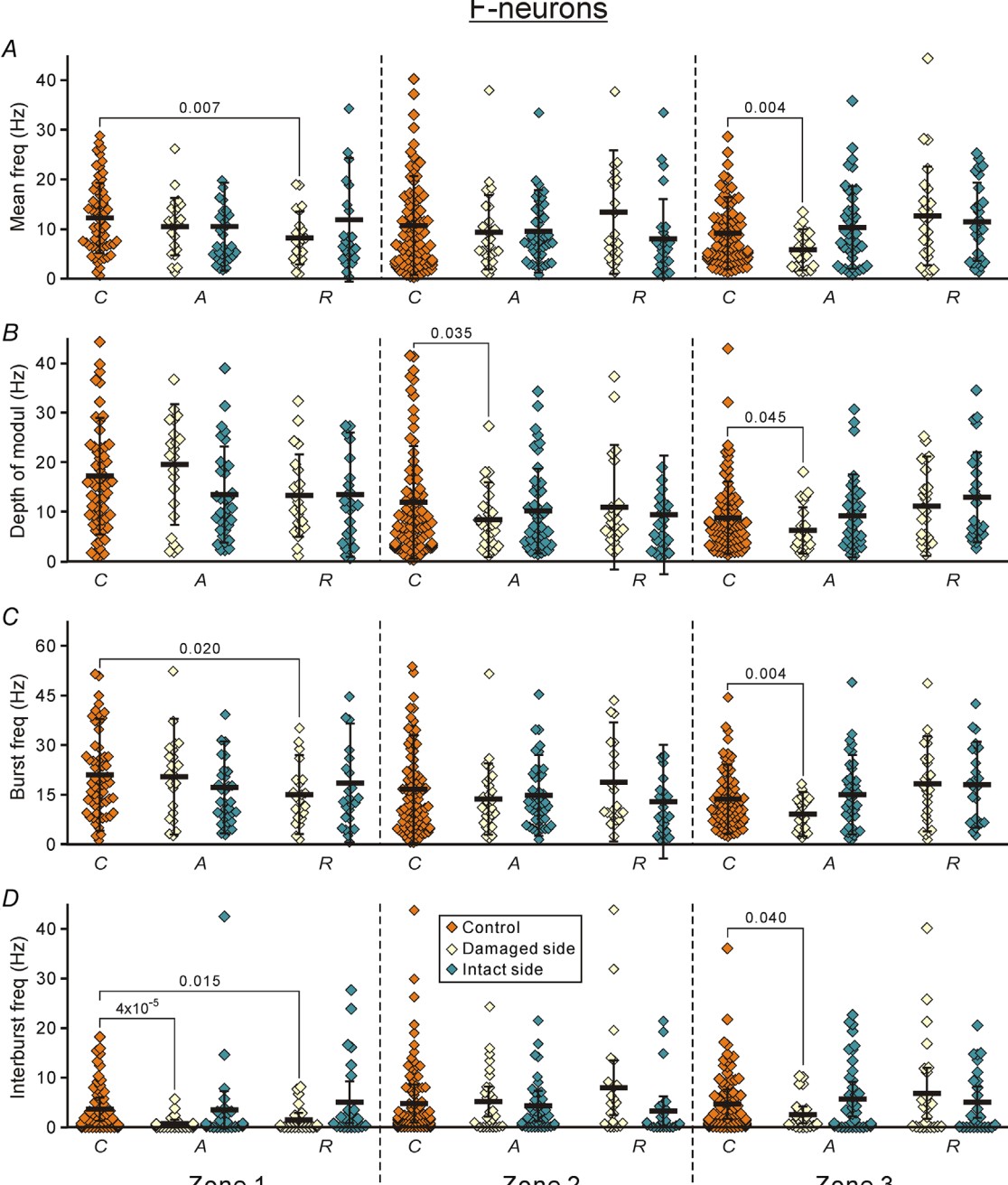

**Figure 4. Comparison of activity characteristics of F-neurons during tilts of the whole platform in R-rabbits with those observed in the control and after acute LHS**

*A–D*, the mean ± SD values of the mean frequency (*A*), the depth of modulation (*B*), the burst frequency (*C*) and the interburst frequency (*D*) of F-neurons recorded in rabbits with an intact spinal cord (Control, *C*), in rabbits with acute LHS (*A*) and R-rabbits (*R*) on the ipsi-LHS and on the co-LHS side. These values (indicated by black lines across corresponding clouds of diamonds) are shown for the subpopulations located in different zones (1–3) of the grey matter (Fig. 3*A* and *B*). The number of animals and the number of F-neurons recorded in zones 1−3 in the control: $N = 20$, $n = 62$, 94 and 93, respectively. The number of animals and the number of F-neurons recorded in zones 1−3 on the ipsi-LHS and co-LHS side after acute LHS: $N = 7$, $n = 23$, 29 and 23, and $n = 34$, 46 and 42, respectively; in R-rabbits: $N = 8$, $n = 22$, 24 and 25, and $n = 25$, 26 and 26, respectively. The *P* values are indicated when there is a statistically significant difference. [Colour figure can be viewed at wileyonlinelibrary.com]

## E-neurons

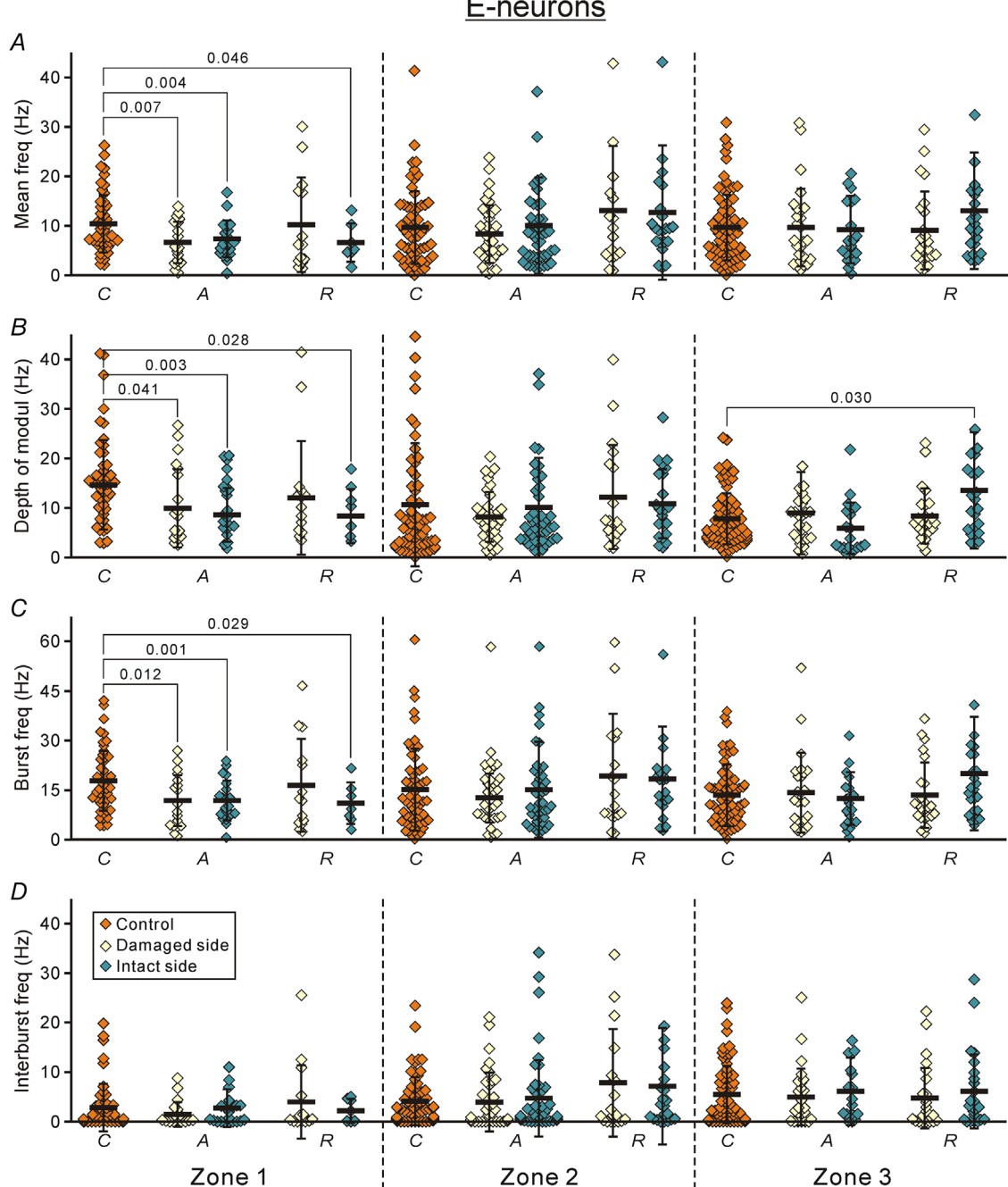

**Figure 5. Comparison of activity characteristics of E-neurons during tilts of the whole platform in R-rabbits with those observed in the control and after acute LHS**

*A–D*, mean ± SD values of the mean frequency (*A*), the depth of modulation (*B*), the burst frequency (*C*) and the interburst frequency (*D*) of E-neurons recorded in rabbits with an intact spinal cord, in rabbits with acute LHS and R-rabbits on the ipsi-LHS and on the co-LHS side of the spinal cord are shown for the subpopulations located in different zones (1–3) of the grey matter. The number of animals and the number of E-neurons recorded in zones 1–3 in the control: *N* = 20, *n* = 49, 63 and 74, respectively. The number of animals and the number of E-neurons recorded in zones 1–3 on the ipsi-LHS and co-LHS side after acute LHS: *N* = 7, *n* = 18, 36 and 26, and *n* = 22, 45 and 20, respectively; in R-rabbits: *N* = 8, *n* = 14, 18 and 24 and *n* = 7, 22 and 24, respectively. Abbreviations as in Fig. 4. [Colour figure can be viewed at wileyonlinelibrary.com]

3, where the depth of modulation was significantly lower and higher compared to the control, respectively; compare *Ipsi-LHS side* in Fig. 6*C* and *D*, *G* and *H*). Changes in the activity parameters of F-neurons located in areas, in which they were significantly changed after acute LHS compared to the control, were different. Thus, in the

areas of the dorsal and ventral horn, where the mean frequency was significantly decreased after acute LHS, it was decreased and increased, respectively, compared to the control (compare Ipsi-LHS side in Fig. 6*A* and *B*). The value of the depth of modulation in the areas of the dorsal and ventral horn, in which it was significantly changed

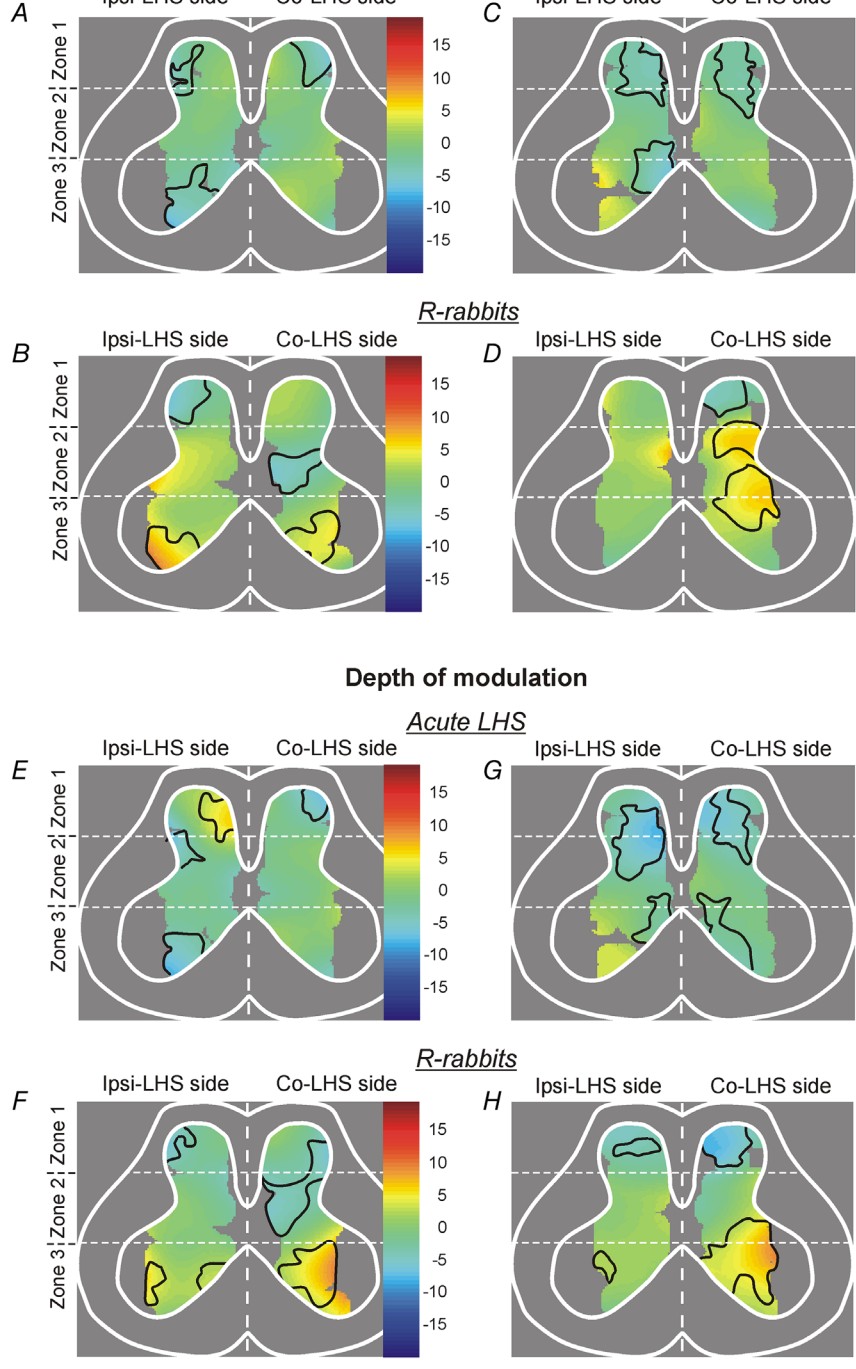

**Figure 6. Changes in the mean frequency and in the depth of modulation of local populations of F- and E-neurons in R-rabbits**

The difference between the averaged distribution of the mean frequency (*A–D*) and of the depth of modulation (*E–H*) of F-neurons (*A* and *B*, and *E* and *F*) and E-neurons (*C* and *F*, and *G* and *H*) on the ipsi-LHS and co-LHS side of the cross-section of the spinal cord in rabbits with acute LHS (*A* and *C*, and *E* and *G*) and in R-rabbits (*B* and *D*, and *F* and *H*) and the corresponding distribution in the control (subtraction of Control from corresponding LHS). The areas of significant changes of the local mean frequency and the local depth of modulation, respectively, are delimited by a solid black line (*t* test, $P < 0.05$). The number of animals and the number of F- and E-neurons recorded in the control: $N = 20$, $n = 249$ and 186, respectively. The number of animals and the number of F- and E-neurons recorded after acute LHS: $N = 7$, respectively, $n = 75$ and $n = 80$ on the ipsi-LHS side, $n = 122$ and $n = 87$ on the co-LHS side; in R-rabbits: $N = 8$, respectively, $n = 71$ and $n = 56$ on the ipsi-LHS side, $n = 77$ and $n = 53$ on the co-LHS side. [Colour figure can be viewed at wileyonlinelibrary.com]

after acute LHS, returned to the control level (compare Ipsi-LHS side in Fig. 6*E* and *F*). In addition, in three small areas (the dorsolateral area in the dorsal horn and the lateral and medial areas in the ventral horn), where the value of the depth of modulation was not affected by acute LHS, it was significantly decreased in the dorsal horn area and significantly increased in the ventral horn areas compared to the control (compare Ipsi-LHS side in Fig. 6*E* and *F*).

By contrast, on the co-LHS side in R-rabbits, significant activity changes in F- and E-neurons were observed not only in areas that were affected by acute LHS, but also in rather large areas that were not affected by acute LHS (compare Co-LHS side in Fig. 6*A* and *B*, *C* and *D*, *E* and *F*, and *G* and *H*). Thus, large areas in zone 2 and zone 3 that contained F-neurons exhibiting a significant decrease and increase, respectively, in both the mean frequency and the depth of modulation compared to the control were not affected by acute LHS (compare Co-LHS side in Fig. 6*B* and *A*, *F* and *E*, respectively). Similarly, E-neurons located in the ventrolateral area of zone 2 and the dorsal area of zone 3, which were not affected by acute LHS, exhibited a significant increase in both the mean frequency and the depth of modulation compared to the control (compare Co-LHS side in Fig. 6*D* and *C*, *H* and *G*, respectively). Among the areas of grey matter in which a significant decrease in activity of F- or E-neurons was observed after acute LHS (Co-LHS side in Fig. 6*A*, *C*, *E* and *G*), in some areas, the corresponding parameter was similar to that in the control (e.g. the mean frequency of F-neurons in the area located in the dorsal horn; Co-LHS side in Fig. 6*B*); in some areas, it was significantly higher (e.g. the mean frequency of E-neurons located in the dorso-lateral part of zone 2 and the depth of modulation of E-neurons in the medial part of zone 3; Co-LHS side in Fig. 6 *D* and *H*, respectively) or still lower than in the control (e.g. the mean frequency and depth of modulation of E-neurons in the area located in the dorsal horn; Co-LHS side in Fig. 6*D* and *H*, respectively).

Thus, in R-rabbits among the areas of grey matter where the activity of F- or E-neurons was significantly decreased after acute LHS, it reached the control level in some areas and did not reach the control level in others. Finally, in some zones where the activity was not affected by acute LHS, it was significantly changed compared to that in the control.

### Changes in balance between the responses to the tilts of the F-neurons population and E-neurons population in LHS-rabbits

In rabbits after acute LHS on the ipsi-LHS side, as well as in R-rabbits on the co-LHS side, a substantial proportion of incorrect EMG responses to platform tilts was observed

(Fig. 2*C*). We hypothesized that the appearance of incorrect EMG responses could be a result of distortion in balance between the response of the F-neuron population and response of the E-neuron population to platform tilt. Because the response of a neuron to the platform tilt is characterized by the depth of modulation, the response of the F-neuron population ($R_F$) located in a particular zone of the grey matter was calculated as $R_F = M_F \times P_F$, where $M_F$ is a mean of the modulation depth of F-neurons recorded in the zone and $P_F$ is a proportion of F-neurons in the population of modulated neurons recorded in this zone. The response of the population of E-neurons ($R_E$) located in a particular zone was calculated correspondingly. Then, to reveal a change in the balance between the response of the F-neuron population and the response of the E-neuron population located in each of the three zones, the difference between $R_F$ and $R_E$ in each of the three zones was calculated for the control, for the ipsi-LHS and co-LHS sides after acute LHS, as well as for the ipsi-LHS and co-LHS sides in R-rabbits. For LHS-rabbits, these differences, which were normalized to the corresponding difference in the control, are shown for each of the three zones in Fig. 7*A*. In the control, in each of the three zones, the value of the difference was positive (indicating a predominance of the F-neuron population response over that of the E-neuron population response) and is considered in Fig. 7*A* as 100% for each of the three zones. One can see that, in rabbits after acute LHS as well as in R-rabbits, there was a strong asymmetry in the differences on the co-LHS and ipsi-LHS sides in zones 2 and 3. Thus, after acute LHS, on the ipsi-LHS side (where incorrect EMG responses dominated) (Fig. 2*C*), the differences had negative values indicating that the response of the E-neuron population dominated over that of the F-neuron population. In R-rabbits, on the co-LHS side (where a substantial proportion of incorrect EMG responses was observed) (Fig. 2*C*), the difference was close to zero. By contrast, on the co-LHS side after acute LHS and on the ipsi-LHS side in R-rabbits (where correct EMG responses dominated) (Fig. 2*C*), the differences had positive values, indicating the predominance of the F-neuron population response over that of the E-neuron population.

In Fig. 7*B*, the normalized difference between the responses of F-population and E-population located in zones 2 and 3 in the control, on the ipsi-LHS and co-LHS sides after acute LHS and on the ipsi-LHS and co-LHS sides in R-rabbits, is plotted against the corresponding difference between the percentage of correct and incorrect EMG responses to platform tilts. We found a strong positive correlation between these parameters, suggesting that a shift in the balance between the responses of F- and E-populations located in zones 2 and 3 from the predominance of the F-population response toward the predominance of the E-population response leads to an

increase in the proportion of incorrect EMG responses and in a decrease in the proportion of correct EMG responses to tilts. By contrast, for the F- and E-populations located in zone 1, such a correlation was absent (Fig. 7*C*).

## Processing of tilt-related sensory information in R-rabbits

**Sources of modulation of F- and E-neurons.** The tilt-related modulation of neuronal activity is determined by somatosensory inputs from limbs. To characterize the changes in the processing of tilt-related sensory information in R-rabbits, in F-neurons and E-neurons recorded on the ipsi-LHS side ($n = 54$ and 46, respectively)

(Table 1) and on the co-LHS side ($n = 61$ and 41, respectively) (Table 1) in addition to responses to the whole platform tilts, we recorded responses to the tilts of the right and left parts of the platform (Fig. 1*G*). The results were compared with the control and with those obtained after acute LHS.

As in the control (Zelenin et al. 2015) and after acute LHS (Zelenin, Lyalka, Orlovsky et al. 2016), in R-rabbits, four types of neurons were found on the ipsi-LHS, as well as on the co-LHS side of the spinal cord. Type 1 and 2 neurons received tilt-related sensory input from the ipsilateral limb only and from the contralateral limb only, respectively. By contrast, Type 3 and 4 neurons received sensory inputs from both hindlimbs. However, Type 3

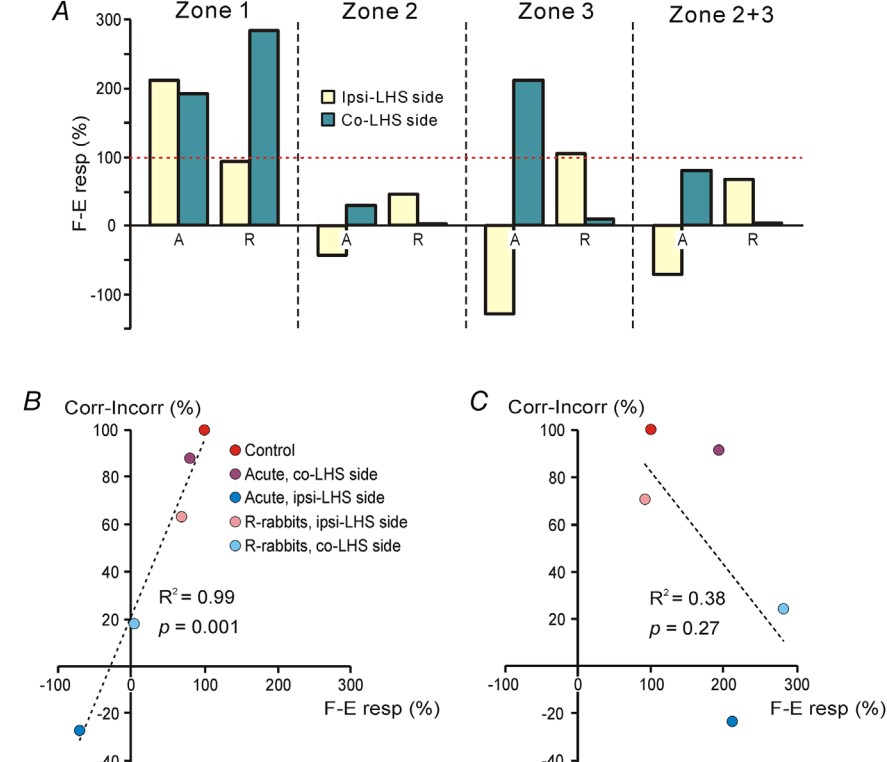

**Figure 7. Changes in balance between responses to tilt of F-neurons population and E-neurons population in LHS-rabbits**

*A*, for rabbits after acute LHS and for R-rabbits, the differences between responses of F-population and E-population on the ipsi-LHS and co-LHS sides, which were normalized to the corresponding difference in the control are shown for each of three zones as well as for zones 2 and 3 together. The red dotted line indicates the value of difference in the control. *B*, a decrease in difference between responses to tilts of F-neurons population and E-neurons population, located in zones 2 and 3, positively correlates with an increase in relative number of incorrect EMG responses to tilts. *C*, difference between responses to tilt of F-neurons population and E-neurons population located in zone 1 does not correlate with the type of EMG response to tilt. In (*B*) and (*C*), the *x*-axis is normalized to the control difference between the response of F-neurons population and E-neurons population to tilts; the *y*-axis is the difference between the percent of correct and incorrect responses to tilts. The number of animals, as well as the number of F- and E-neurons recorded in zones 1−3 in the control: $N = 20$, $n = 62$, 94 and 93, and $n = 49$, 63 and 74, respectively. The number of animals as well as the number of F- and E-neurons recorded in zones 1−3 after acute LHS: $N = 7$, respectively $n = 23$, 29 and 23, and $n = 18$, 36 and 26 on the ipsi-LHS side, $n = 34$, 46 and 42 and $n = 22$, 45 and 20 on the co-LHS side; in R-rabbits: $N = 8$, respectively, $n = 22$, 24 and 25, and $n = 14$, 18 and 24 on the ipsi-LHS side, $n = 25$, 26 and 26, and $n = 7$, 22 and 24 on the co-LHS side. Abbreviations as in Fig. 4. [Colour figure can be viewed at wileyonlinelibrary.com]

neurons were activated by ipsilateral limb flexion and contralateral limb extension or ipsilateral limb extension and contralateral limb flexion, whereas Type 4 neurons were activated by the flexion of each of the hindlimbs or by the extension of each of the hindlimbs.

As after acute LHS, the relative number of Type 1 F- and E-neurons on both the ipsi-LHS and co-LHS sides in R-rabbits was approximately twofold higher than it was in the control (Figs 8*A* and 9*A*). Correspondingly, the proportion of neurons with input from the contralateral hindlimb (Types 2−4) decreased by about two-fold on both sides of the spinal cord. These changes in the proportions of Type 1 neurons and Type 2−4 neurons were significant for both the F- and E-groups on both the ipsi-LHS and co-LHS sides in R-rabbits (chi-squared test for the number of neurons receiving or not receiving

inputs from the contralateral limb, in the control and R-rabbits, $P = 2 \times 10^{-6}$ and $2 \times 10^{-7}$ for F- and E-neurons on the ipsi-LHS side, respectively; $P = 10^{-6}$ and $3 \times 10^{-4}$ for F- and E-neurons on the co-LHS side, respectively). The proportion of Type 1 F- and E-neurons on both the ipsi-LHS and co-LHS sides in R-rabbits was similar to that observed after acute LHS (chi-squared test, $P = 0.383$ and 0.748 for F- and E-neurons on the ipsi-LHS side, respectively; $P = 0.878$ and 0.148 for F- and E-neurons on the co-LHS side, respectively). These results suggest that sensory input from the contralateral limb to F- and E-neurons does not recover in R-rabbits.

**Efficacy of sensory inputs to F- and E-neurons from different limbs.** F- and E-neurons are modulated by tilt-related somatosensory inputs from the limbs. The

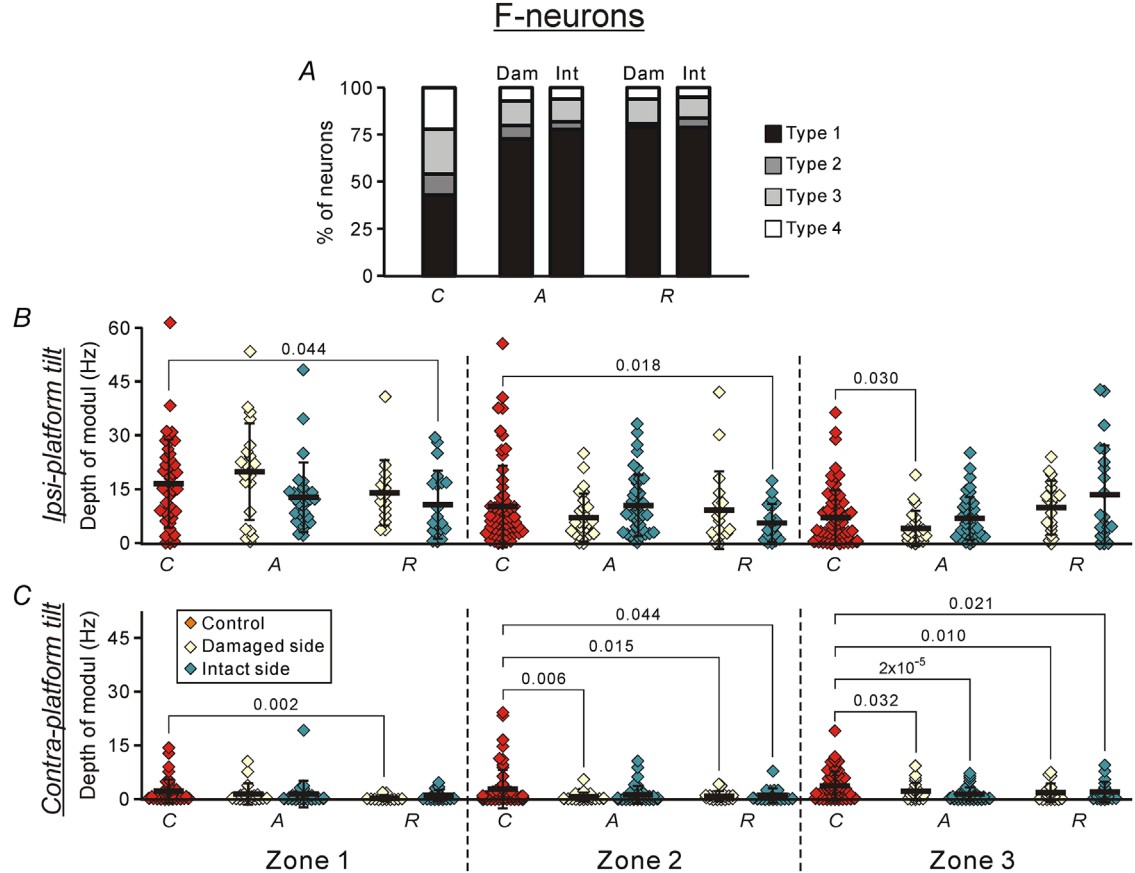

**Figure 8. Changes in tilt-related sensory signals to F-neurons in R-rabbits**
*A*, percentage of F-neurons receiving different combinations of tilt-related somatosensory inputs from the limbs (Types 1−4) in the control, after acute LHS and in R-rabbits. For explanation, see text. *Dam* and *Int*, the ipsi-LHS and co-LHS side of the spinal cord, respectively. *B* and *C*, mean ± SD values of the depth of modulation of F-neurons caused by ipsilateral platform tilt (*B*) and by contralateral platform tilt (*C*) in the control rabbits, as well as on the ipsi-LHS and co-LHS side in rabbits after acute LHS and in R-rabbits. Abbreviations as in Fig. 4. The number of animals and the number of F-neurons from zones 1, 2 and 3 for the control: $N = 20$, $n = 42$, 62 and 71; for acute LHS: $N = 7$, $n = 21$, 27 and 22 on the ipsi-LHS side and $n = 30$, 41 and 41 on the co-LHS side; for R-rabbits; $N = 8$, $n = 15$, 19 and 20 on the ipsi-LHS side and $n = 20$, 20 and 21 on the co-LHS side. The *P* values are indicated when there is a statistically significant difference. [Colour figure can be viewed at wileyonlinelibrary.com]

efficacy of sensory input from a particular limb to a neuron is reflected in its activity modulation depth caused by the tilts of the platform under the limb. In R-rabbits, on the ipsi-LHS side, the efficacy of the sensory input from the ipsilateral limb to F-neurons located in zone 3 (which was significantly decreased after acute LHS) (Fig. 8*B*) recovered to the control level (Welch's *t test*, $P = 0.117$) (Fig. 8*B*). By contrast, on the co-LHS side, the efficacy of sensory input to F-neurons was significantly decreased in zones 1 and 2 (Fig. 8*B*) where it was unaffected by acute LHS (Welch's *t* test, $P = 0.126$ and 0.905, respectively) (Fig. 8*B*). The efficacy of the sensory input from the ipsilateral limb to the E-neurons (which was not affected by acute LHS in any of the three zones; Welch's *t* test, for the ipsi-LHS side: $P = 0.949$, 0.509, and 0.682 in zones 1, 2 and 3, respectively; for the co-LHS side: $P = 0.126$, 0.875 and 0.158 in zones 1, 2 and 3, respectively) remained similar to the control in R-rabbits (Welch's *t*

test, for the ipsi-LHS side: $P = 0.852$, 0.467 and 0.507 in zones 1, 2 and 3, respectively; for the co-LHS side: $P = 0.341$, 0.506 and 0.193 in zones 1, 2 and 3, respectively) (Fig. 9*B*).

The efficacy of the sensory input from the contralateral limb to F-neurons was significantly decreased compared to the control on the ipsi-LHS side in zone 1 (Fig. 8*C*) (where it was not affected by acute LHS; Welch's *t* test, $P = 0.437$) and on both sides in zones 2 and 3 (Fig. 8*C*), where, after acute LHS, it was significantly decreased on the ipsi-LHS side in zone 2 (Fig. 8*C*) and on both sides in zone 3 (Fig. 8*C*). The sensory input from the contralateral limb to E-neurons, which was significantly decreased on both sides in each of the three zones after acute LHS (Fig. 9*C*), did not differ from control only on the ipsi-LHS side in zone 1 (Welch's *t* test, $P = 0.081$) and on the co-LHS side in zones 2 and 3 (Welch's *t* test, $P = 0.090$ and 0.155, respectively).

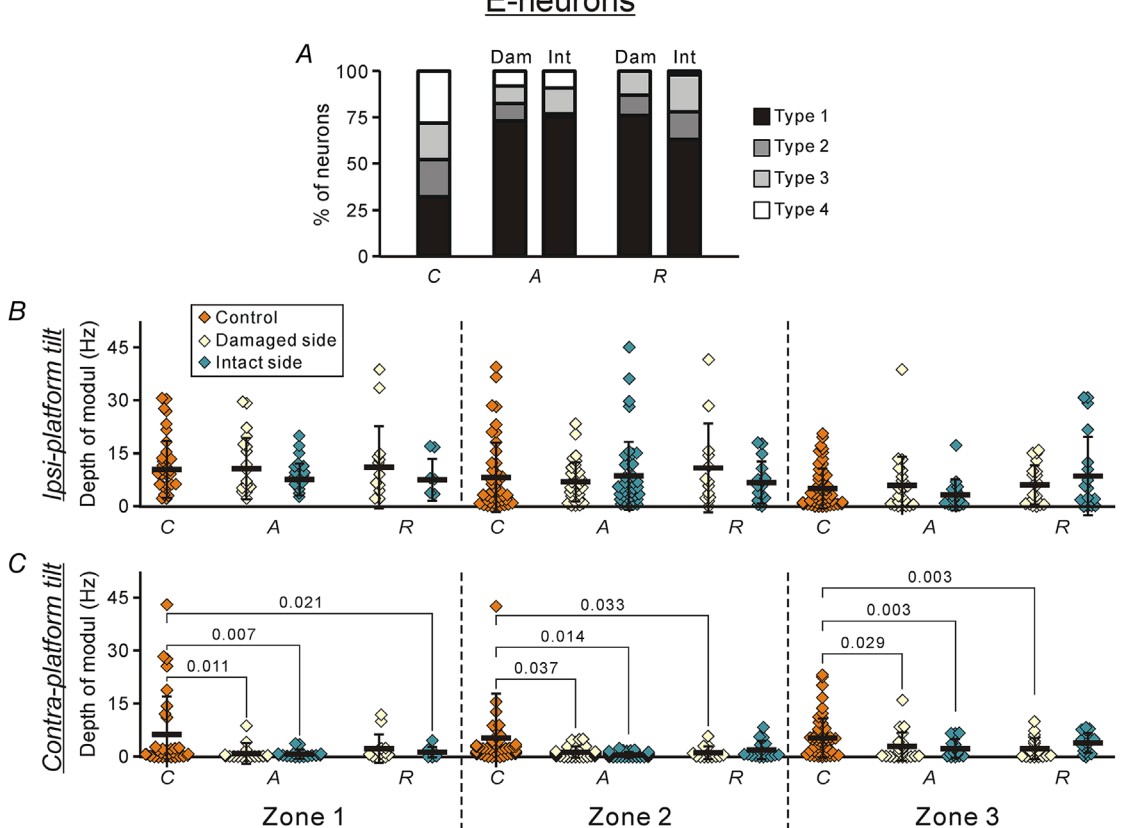

**Figure 9. Changes in tilt-related sensory signals to E-neurons in R-rabbits**
*A*, percentage of E-neurons receiving different combinations of tilt-related somatosensory inputs from the limbs (Types 1−4) in the control, after acute LHS and in R-rabbits. For explanation, see text. *B* and *C*, mean ± SD values of the depth of modulation of E-neurons caused by ipsilateral platform tilt (*B*) and by contralateral platform tilt (*C*) in the control rabbits, as well as on the ipsi-LHS and co-LHS side in rabbits after acute LHS and in R-rabbits. Abbreviations as in Figs 4 and 8. The number of animals and the number of E-neurons from zones 1, 2 and 3 for the control: $N = 20$, $n = 32$, 45 and 55; for acute LHS: $N = 7$, $n = 17$, 33 and 24 on the ipsi-LHS side and $n = 19$, 43 and 17 on the co-LHS side; for R-rabbits; $N = 8$, $n = 13$, 13 and 20 on the ipsi-LHS side and $n = 6$, 16 and 19 on the co-LHS side. The *P* values are indicated when there is a statistically significant difference. [Colour figure can be viewed at wileyonlinelibrary.com]

**Relation between responses to tilts and receptive fields of neurons.** In R-rabbits, somatosensory receptive fields were found in 54 out of 73 tested modulated neurons on the ipsi-LHS side and in 51 out of 74 on the co-LHS side. In R-rabbits, the proportion of neurons with receptive fields on both ipsi-LHS and co-LHS sides of the spinal cord was significantly decreased compared to the control (chi-squared test, $P = 0.010$ and $4 \times 10^{-4}$, respectively) and similar to that in rabbits after acute LHS (chi-squared test, $P = 0.475$ and $0.057$, respectively). As in the control and after acute LHS, in R-rabbits, in the majority of neurons on the ipsi-LHS and co-LHS sides, the receptive fields were 'deep': the neurons responded to the palpation of muscles or to the movements of joints but not to the stimulation of fur or skin alone (Fig. 10*A*). The percentage of neurons with multiple (from more than one muscle) deep receptive fields in R-rabbits on the ipsi-LHS side did not differ from that in the control and after acute LHS (chi-squared test, $P = 0.379$ and $0.071$, respectively), whereas, on the co-LHS side, it was similar to the control (chi-squared test, $P = 0.241$) but significantly lower than after acute LHS (chi-squared test, $P = 0.012$). By contrast, in R-rabbits, the relative number of neurons with a receptive field from one muscle was significantly reduced compared to the control on both the ipsi-LHS and co-LHS sides (chi-squared test, $P = 0.045$ and $0.012$, respectively) and similar to that after acute LHS (chi-squared test, $P = 0.408$ and $0.545$, respectively). Finally, a small proportion of neurons with sensory input from the skin in R-rabbits (which was almost absent after acute LHS) did not differ significantly from that in the control (chi-squared test, $P = 0.081$ and $0.088$ on the ipsi-LHS and co-LHS sides, respectively). Thus, in general, the receptive fields of modulated neurons did not return to the control state in R-rabbits. Changes in the receptive

fields of dorsal horn neurons in chronically hemisected animals were demonstrated previously (Brenowitz & Pubols, 1981).

For modulated neurons with deep receptive fields located on the ipsi-LHS and co-LHS sides ($n = 27$ and $21$, respectively, in R-rabbits), we compared the responses of a neuron to tilts with afferent signals that the neuron presumably receives from its receptive field during tilts. One could expect that a tilt of the platform would activate stretch and load receptors in the extensors of the flexing limb, as well as those in the flexors of the extending limb.

In R-rabbits, as in the control and after acute LHS, three types of neurons were found on both the co-LHS and ipsi-LHS sides: (i) neurons with responses to tilts that could be fully explained by receptive field inputs, such as an F-neuron with excitatory inputs only from one or several extensor muscles (*Expl* in Fig. 10*B*); (ii) neurons with afferent inputs that could be responsible for their reaction to tilts, although they also had mismatching inputs, such as excitatory inputs from the antagonistic muscles of one limb (*Partly expl* in Fig. 10*B*); (iii) and neurons that modulation could not be explained by input from the receptive field (*Not expl* in Fig. 10*B*).

In R-rabbits, similar to after acute LHS, an increase that is more than two-fold in the relative number of neurons on both the ipsi-LHS and co-LHS sides in which the receptive field input could explain the response to tilts (56% and 67%, respectively, in R-rabbits *vs.* 23% in the control; chi-squared test, $P = 4 \times 10^{-4}$ and $2 \times 10^{-5}$) and the corresponding decrease in the relative number of neurons in which the response could not be explained by input from the receptive field (22% and 9% in R-rabbits *vs.* 48% in the control; chi-squared test, $P = 0.012$ for the ipsi-LHS side, $P = 7 \times 10^{-4}$ for the co-LHS side) was observed (Fig. 10*B*).

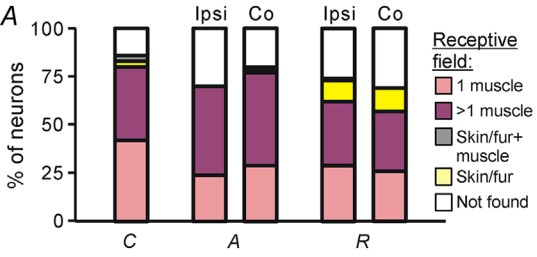
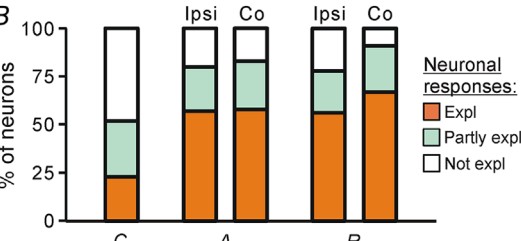

**Figure 10. Changes in receptive fields of F- and E-neurons in R-rabbits**

*A*, proportion of neurons receiving sensory inputs from different sources; i.e. from receptors of only one muscle (*1 muscle*), from receptors of more than one muscle (*>1 muscle*), from cutaneous and muscle receptors (*Skin/fur+muscle*), from cutaneous receptors only (*Skin/fur*) and with no receptive field found (*Not found*) in the control, after acute LHS and in R-rabbits. For explanation, see text. The number of animals and the number of neurons for the control: $N = 20$, $n = 286$; for acute LHS: $N = 7$, $n = 140$ on the ipsi-LHS side and $n = 180$ on the co-LHS side; for R-rabbits; $N = 8$, $n = 73$ on the ipsi-LHS side and $n = 74$ on the co-LHS side. *B*, proportion of neurons in which response to tilts could be completely explained (*Expl*), partly explained (*Partly expl*) and could not be explained (*Not expl*) by input from their receptive field, in the control, after acute LHS and in R-rabbits. The number of animals and the number of neurons for the control: $N = 20$, $n = 211$; for acute LHS: $N = 7$, $n = 75$ on the ipsi-LHS side and $n = 114$ on the co-LHS side; for R-rabbits; $N = 8$, $n = 27$ on the ipsi-LHS side and $n = 21$ on the co-LHS side. [Colour figure can be viewed at wileyonlinelibrary.com]

## Discussion

In the present study, we characterized the activity of spinal interneurons of postural networks, as well as the processing of posture-related sensory signals in rabbits with postural functions recovered after LHS (R-rabbits). A comparison of these data with the data obtained in our previous studies on rabbits with intact spinal cord (control; Zelenin et al. 2015) and on rabbits after acute LHS (Zelenin, Lyalka, Orlovsky et al. 2016) allowed us to characterize the changes in the spinal postural networks underlying the recovery of postural functions. We demonstrate that, in R-rabbits, plastic changes take place not only in postural networks for which the operation was dramatically distorted by acute LHS, but also in the postural networks that were almost not affected by acute LHS.

Changes in postural networks below the forebrain underlie the recovery of postural corrections on the ipsi-LHS side, whereas changes in the forebrain substantially contribute to the generation of postural corrections on the co-LHS side.

We have shown that, in R-rabbits, the EMG pattern (in terms of the phases of EMG responses in the tilt cycle) and the efficacy of postural corrections in the hindquarters (reflected in their gain) are similar to those observed in the control. PLRs are an important component of limb corrective movements caused by the disturbance of body orientation in the transverse plane (Musienko et al., 2008, 2010; Deliagina et al., 2012, 2014). We have demonstrated that the impairments of postural functions caused by acute LHS or the complete transection of the spinal cord are well reflected in the PLR pattern (Zelenin et al. 2013; Zelenin, Lyalka, Hsu et al., 2016; Zelenin, Lyalka, Orlovsky et al. 2016). One can expect that the recovery of postural functions is also well reflected in the PLR pattern and that PLRs in R-rabbits are similar to those in the control on both the ipsi-LHS and co-LHS sides. As expected, PLRs on the ipsi-LHS side, where they were absent after acute LHS (Zelenin, Lyalka, Orlovsky et al. 2016), were similar to the control in R-rabbits. The recovery of PLRs on the ipsi-LHS side observed in decerebrate R-rabbits suggests that this was caused by changes in the postural networks residing below the decerebration level (in the brainstem–cerebellum–spinal cord).

A number of factors could contribute to the recovery of PLRs on the ipsi-LHS side after LHS. First, an increase in the excitability of spinal neurons for which the activity was significantly decreased after acute LHS (Zelenin, Lyalka, Orlovsky et al. 2016). Two mechanisms can contribute to this: (i) the re-innervation of spinal neurons located below the lesion by supraspinal and propriospinal neurons with axons running on the co-LHS side of the spinal cord (Bareyre et al. 2004; Vavrek et al. 2006; Fenrich & Rose, 2009; Flynn et al. 2011;

Oudega & Perez, 2012; Zörner et al. 2014) and (ii) the development of denervation supersensitivity and neuronal hyperexcitability (Cannon, 1932; Thesleff & Sellin, 1980). Previously, we demonstrated that spinalization triggers two processes of plastic changes in the spinal postural networks: the rapid restoration (taking days) of normal activity level in spinal interneurons and the slow recovery (taking months) of motoneuronal excitability (Zelenin et al. 2019). Furthermore, Thaweerattanasinp et al. (2020) demonstrated an increase in excitability of a subset of deep dorsal horn neurons correlating with increased excitability of motoneurons in chronic spinal mice and obtained evidence that these changes were related to alterations in serotonin input from the brainstem.

The second factor that could contribute to the recovery of PLRs on the ipsi-LHS side after LHS is a recovery of efficacy of sensory input from limb mechanoreceptors (which was decreased after LHS; Zelenin, Lyalka, Orlovsky et al. 2016) as a result of the restoration of the normal processing of this input. The third factor could be a recovery of the sensory input efficacy from the limb. The latter could be caused by the recovery of the excitability of extensor motoneurons leading to the appearance of forces developed by extensor muscles on the ipsi-LHS side (Fig. 2*I*) (which were absent after acute LHS; Zelenin, Lyalka, Orlovsky et al. 2016) and monitored by load receptors. It is also possible that the recovery of gamma-motoneurons excitability [which receive significant supraspinal influences in the control (Pompeiano, 1972; Granit, 1979; Hulliger, 1984) and thus most probably their excitability after acute LHS was substantially reduced], leads to an increase in signals from muscle spindles. The present study has demonstrated that the recovery of activity of F- and E-neurons, as well as the efficacy of sensory input from limbs in most areas of grey matter, where they were significantly decreased after acute LHS, are among these factors.

Surprisingly, we found that, in decerebrate R-rabbits, the phases of EMG responses in the tilt cycle during PLRs were substantially distorted on the co-LHS side, where they were similar to the control after acute LHS (Zelenin, Lyalka, Orlovsky et al. 2016). On the co-LHS side in R-rabbits, correct EMG responses to tilts were observed only in 42% of the cycles, whereas, in 27% and 31% of cycles, EMG responses were incorrect and absent, respectively. Weak residual incorrect responses to tilts were also observed on the ipsi-LHS side after acute LHS (Zelenin, Lyalka, Orlovsky et al. 2016) and after acute spinalization (Zelenin, Lyalka, Hsu et al., 2016). Because experiments with reversible LHS (Zelenin, Lyalka, Orlovsky et al. 2016), as well as with reversible spinalization (Zelenin et al. 2013), demonstrated that a small proportion of non-modulated neurons became F- and E-neurons, it was suggested that these neurons are

responsible for the appearance of residual incorrectly phased EMG responses to tilts (Zelenin, Lyalka, Orlovsky et al. 2016) and a hypothesis explaining the mechanism underlying the appearance of these incorrect EMG responses has been proposed (Zelenin, Lyalka, Orlovsky et al. 2016). A possible mechanism underlying the distortion and recovery of PLRs on the co-LHS and ipsi-LHS sides, respectively, in R-rabbits is described in the framework of this hypothesis below.

Considering that the EMG pattern of postural corrections on the co-LHS side in R-rabbits was similar to the control and that PLRs substantially contribute to postural corrections, the distortion of PLRs on the co-LHS side in R-rabbits observed after decerebration suggests that the forebrain most probably contributes substantially to the generation of postural corrections on the co-LHS side in R-rabbits. Previously, we demonstrated that the integrity of the forebrain is not necessary for the generation of postural corrections in response to platform tilts (Musienko et al. 2008). Recording the activity of the corticospinal neurons during postural corrections caused by platform tilts in intact subjects led to the suggestion that its functional role is to increase the excitability level of the basic postural networks residing at the brainstem–cerebellum–spinal levels (Beloozerova et al. 2005). One can expect that, in R-rabbits, the activity of corticospinal neurons during postural corrections is substantially modified compared to the control and that corticospinal signals are important for the generation of the EMG pattern of postural corrections. Thus, in R-rabbits, plastic changes take place in both postural networks in which the operation was distorted and postural networks that were almost not affected by acute LHS.

It is demonstrated that, after LHS, plastic changes develop at different CNS levels. Thus, the reorganization of limb representation in the primary motor cortex on the lesioned side (Zörner et al. 2014; Hilton et al., 2016; Brown & Martinez, 2018, 2021), corticospinal axonal plasticity (Friedli et al. 2015; Nishimura & Isa, 2009, 2012; Rosenzweig et al. 2010; Oudega & Perez, 2012), including the formation of new connections with propriospinal neurons located rostrally to the lesion for which the axons descend in white matter spared after LHS (Bareyre et al. 2004; Courtine et al. 2008; Vavrek et al. 2006), the sprouting of uninjured reticulospinal axons to the lesioned side of the spinal cord below the LHS (Zörner et al. 2014; Engmann et al. 2020), as well as the formation of new connections by propriospinal neurons (Fenrich & Rose, 2009; Flynn et al. 2011), contribute to the recovery of motor functions after LHS. Plastic reorganization also occurs in the spinal cord below LHS. The sprouting of dorsal root terminals in spinal laminae paralleling the recovery of motor functions has been reported previously (Bullitt et al. 1988; Helgren & Goldberger, 1993;

Hollis et al. 2016; Murray & Goldberger, 1974). Plastic changes in the central pattern generator and reflex transmission below the lesion are involved in the recovery of locomotion after LHS. These changes result in a clear left–right asymmetry (i.e. the enhancement of spinal reflexes and the capacity to generate locomotion on the ipsi-LHS side compared to those on the co-LHS side) (Barriere et al. 2008; Gossard et al. 2015; Hultborn & Malmsten, 1983a, 1983b; Martinez et al., 2011, 2012). Revealing the relative contribution of plastic changes taking place at different levels of CNS to the recovery of postural functions after LHS is an objective for future studies.

## The proportion of different types of neurons on both the ipsi-LHS and co-LHS sides does not return to the control level

As in the control and after acute LHS, in R-rabbits we found three groups of neurons (F-neurons, E-neurons, and non-modulated neurons) in each of the three zones of grey matter. Because control F- and E-neurons are modulated in-phase and in antiphase with extensor motoneurons, it was suggested that at least some of them are pre-motor interneurons that excite and inhibit extensor motoneurons, respectively (Zelenin et al. 2015).

It was suggested that one of the factors contributing to the disappearance of PLRs and the appearance of residual incorrect EMG responses on the ipsi-LHS side after acute LHS is a significant decrease in the proportion of F-neurons on the ipsi-LHS side (Zelenin, Lyalka, Orlovsky et al. 2016). Experiments with reversible LHS have shown that this decrease can be explained by a predominance of F-neurons in the population of modulated neurons that lost modulation or were completely inactivated by LHS on the ipsi-LHS side (Zelenin, Lyalka, Orlovsky et al. 2016). One can expect that, in R-rabbits, the relative number of F-neurons on the ipsi-LHS side reaches the control level, which would contribute to the recovery of PLRs on this side, whereas the proportion of F-neurons on the co-LHS side decreases, which would contribute to the distortion of PLRs and the appearance of incorrect EMG responses on this side. As expected, on the co-LHS side, the proportion of F-neurons (which was not affected by acute LHS) was significantly decreased. One can assume that, in R-rabbits, as a result of plastic changes, the modulation of a part of the F-neurons on the co-LHS side is determined by phasic supraspinal input driven by the forebrain. This explains the decrease in the proportion of F-neurons and the corresponding increase in the proportion of non-modulated neurons observed after decerebration.

Surprisingly, we found that the proportion of F-neurons on the ipsi-LHS side in R-rabbits did not reach the control level. One of possible explanations could be that,

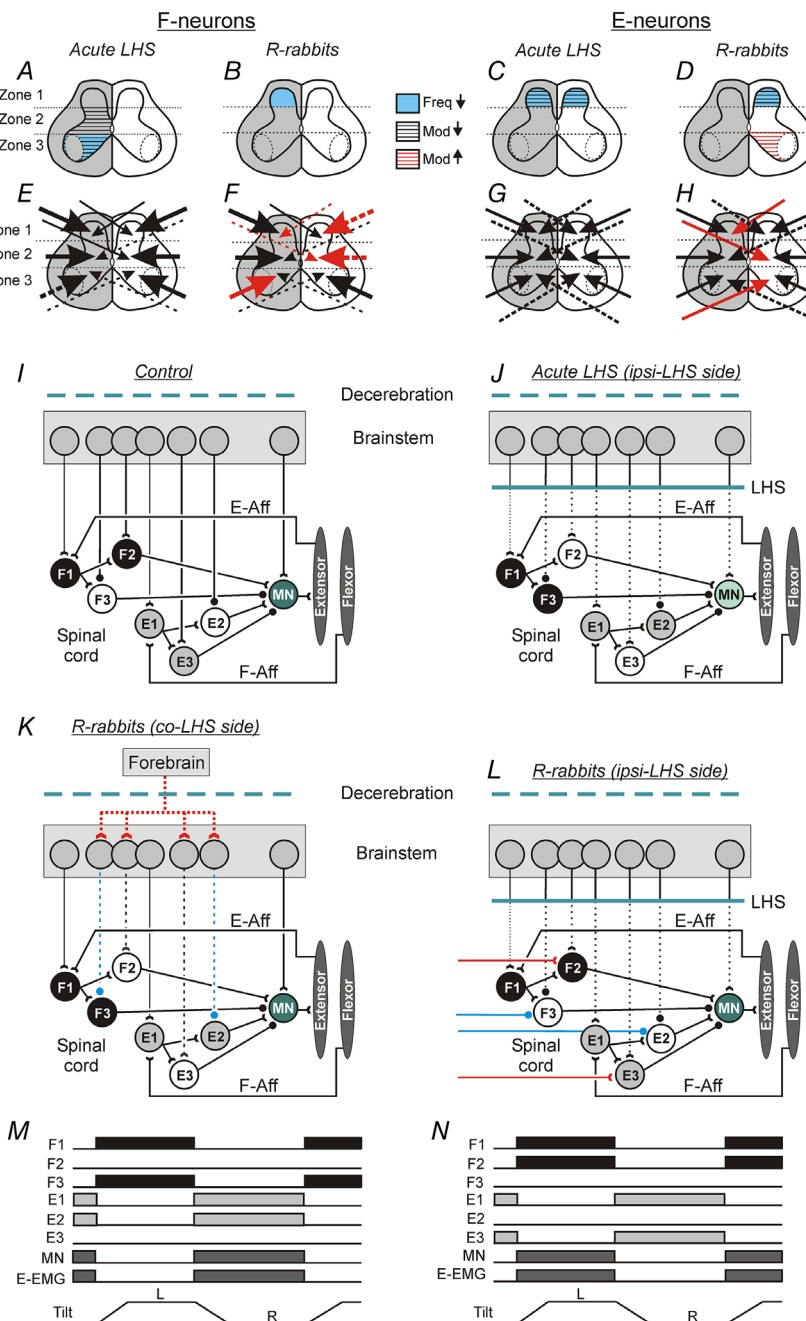

**Figure 11. Summary of results and hypothetical spinal circuitry generating correct and incorrect EMG responses to tilts**

*A–D*, changes (compared to the control) in activity parameters of F-neurons (*A* and *B*) and E-neurons (*C* and *D*) located in three zones of the grey matter after acute LHS (*A* and *C*) and in R-rabbits (*B* and *D*). Shaded half of the spinal cord cross-section indicates the side of LHS. Zones with a significant decrease in the mean frequency are marked by blue. Zones with a significant increase and decrease in the depth of modulation are marked by red and black horizontal hatching, respectively. *E–H*, changes (as compared to the control) in the efficacy of tilt-related sensory inputs from the ipsilateral and from the contralateral limb to F-neurons (*E* and *F*) and E-neurons (*G* and *H*) after acute LHS and in R-rabbits. Arrows from the ipsilateral and contralateral side indicate the tilt-related somatosensory inputs from the ipsilateral and contralateral limbs, respectively. The thickness of the arrows indicates the strength of the input in the control. Black dashed arrows indicate a significant decrease in the efficacy of sensory input as compared to the control observed in both R-rabbits and rabbits after acute LHS. The red solid arrows indicate sensory inputs which efficacy was significantly decreased after acute LHS but recovered to the control level in R-rabbits. Red dashed arrows indicate sensory inputs which efficacy was similar to control after acute LHS but significantly decreased in R-rabbits.

*I–L*, hypothetical spinal circuit generating motor responses to platform tilts in the control (*I*), on the ipsi-LHS side after acute LHS (*L*), as well as on the co-LHS (*K*) and ipsi-LHS (*L*) side in R-rabbits. Only a circuit related to the right limb is shown. The circuit is driven by the tilt-related somatosensory input from the ipsilateral limb. It includes F- and E-groups of neurons; each of the groups consists of three subgroups (F1, F2 and F3, and E1, E2 and E3, respectively). F1 and E1 neurons receive the tilt-related somatosensory input and are activated by flexion and extension of the ipsilateral limb, respectively. F1 and E1 neurons can affect motoneurons of extensor muscles (MNs) of the ipsilateral limb through excitatory (F2 and E2) and inhibitory (F3 and E3) interneurons, respectively. In the control (*I*), F3 and E2 neurons are inhibited, whereas F1, F2, E1, E3 and MNs are excited by supraspinal drive originating from the brainstem and the forebrain does not contribute to generation of EMG pattern of PLRs (Musienko et al. 2008; Hsu et al. 2012). Thus, platform tilts evoke correct EMG responses in extensors: they are activated during ipsilateral limb flexion (through the reflex chain E-Aff→F1→F2→MN) and inactivated during the limb extension (through the reflex chain F-Aff→E1→E3→MN) (*A* and *E*). Acute LHS (*J*) causes (i) an increase in the excitability of F3 and E2 neurons (as a result of disinhibition); (ii) a decrease in the excitability of F2, E3 neurons and MNs (as a result of a loss of excitatory supraspinal drive); and (iii) only small changes in the excitability of E1 and F1 neurons (because the excitatory drive to these neurons is either weak or absent). Inconsistent correct and incorrect motor responses to tilts observed after spinal cord injury can be explained by spontaneous fluctuation of the excitability of F2, F3, E2 and E3 neurons. If the excitability of F3 and E2 neurons is higher than that of F2 and E3 neurons, flexion/extension movements of the limb will activate the reflex chains E-Aff→F1→F3→MN and F-Aff→E1→E2→MN, respectively, resulting in the generation of incorrect responses (*J* and *M*). If the excitability of F2 and E3 neurons is higher than that of F3 and E2 neurons (as in the control; *I*), the correct responses will be generated (*N*). Finally, there will be no responses if the excitability of F2, F3, E2 and E3 is low. *K* and *L*, see explanations in the text. *Designations*: interneurons activated by left tilt, by right tilt and not activated by any tilt are shown in black, grey and white, respectively. MNs with low and high level of excitability are indicated by light and dark green. Open triangles indicate excitatory synapses; small filled circles indicate inhibitory synapses. Continuous and dotted lines indicate, respectively, connections that persisted and were abolished after spinalization or decerebration. The proposed connections could be mono- or polysynaptic. Excitatory and inhibitory connections which appeared or were modified in R-rabbits are indicated by red and blue, respectively. Thick green dashed and solid lines indicate levels of decerebration and LHS, respectively. *M* and *N*, responses of neurons of the circuit to tilts underlying generation of the correct (*M*) and incorrect (*N*) EMG responses. Horizontal bars show the phases of neuronal activity, as well as extensor muscle activity (E-EMG). [Colour figure can be viewed at wileyonlinelibrary.com]

on the ipsi-LHS side, the proportion of non-modulated neurons increased not because a part of F-neurons did not restore their excitability level but because the number of non-modulated neurons increased as a result of an increase in the excitability level of neurons, which were silent in the control and after acute LHS.

### The activity of F- and E-neurons in areas of the spinal cord affected by acute LHS returns to the control level but changes in the areas unaffected by acute LHS

It was suggested that one of the factors contributing to the disappearance of PLRs on the ipsi-LHS side after acute LHS is a significant decrease in the activity of F-neurons in zone 3 on the ipsi-LHS side and E-neurons located on both sides in zone 1 (marked with blue in Fig. 11*A* and *C*, respectively) (Zelenin, Lyalka, Orlovsky et al. 2016). We demonstrated that, in R-rabbits, all activity parameters of F-neurons in zone 3 and of E-neurons in zone 1 on the ipsi-LHS side reached the control value, whereas the activity of E-neurons in zone 1 on the co-LHS side was still significantly decreased compared to the control (Fig. 11*B* and *D*). Most probably, activity recovery in premotor F-neurons in zone 3 largely contributes to the recovery of PLRs on the ipsi-LHS side. The role of F- and E-neurons located in zone 1 in the generation of PLRs is not clear. Thus, although the disappearance of PLRs after acute LHS coincides with a decrease in

E-neuron activity on both sides in zone 1, the recovery of PLRs on the ipsi-LHS side in R-rabbits coincides with the recovery of E-neuron activity in zone 1 of the ipsi-LHS side only and a simultaneous significant decrease in F-neuron activity in the same zone. We found that a shift in the balance between the responses of F- and E-populations located in zones 2 and 3 from the predominance of F-population response toward the predominance of E-population response leads to the distortion of PLRs: an increase in the proportion of incorrect EMG responses and a decrease in the proportion of correct EMG responses to tilts. By contrast, for the F- and E-populations located in zone 1, such a correlation was absent. Thus, the normal activity of F- and E-neurons in zones 2 and 3 (but not in zone 1) is critically important for the generation of a normal EMG pattern of PLRs.

In R-rabbits, we found significant changes in the activity of F- and E-neurons in areas where it was not affected by acute LHS. Thus, in R-rabbits, the activity of F-neurons was significantly decreased on the ipsi-LHS side in zone 1 (marked with blue in Fig. 11*B*) and the depth of modulation of E-neurons was significantly increased on the co-LHS side in zone 3 (marked with red horizontal hatching in Fig. 11*D*), whereas, after acute LHS, these activity parameters in the corresponding zones were similar to control (Fig. 11*A* and *B*). In zone 2 on the co-LHS side, where the activity of the local populations of F- and E-neurons after acute LHS was similar to

the control, in R-rabbits, activity in a local population of F-neurons was significantly decreased, whereas, in a local population of E-neurons, it was significantly increased (compare Fig. 6*B* and *D*). These changes in the activity (especially in the depth of modulation) of local populations of F- and E-neurons located in zone 2, as well as an increase in the depth of modulation of E-neurons in zone 3, contributed to the shift in the balance between the responses of F- and E-populations toward the predominance of E-population response that led to the distortion of PLRs on the co-LHS side. One can suggest that, in R-rabbits, F- and E- neurons located in these areas of grey matter belong to postural networks generating PLRs on the co-LHS side, which are substantially distorted in decerebrate R-rabbits. Most probably, before decerebration, the close-to-normal excitability level of F- and E- neurons in these networks is maintained by forebrain-driven supraspinal signals.

### The processing of tilt-related sensory information from limbs in R-rabbits differs from those after acute LHS and in the control

We demonstrated that, in R-rabbits, the efficacy of sensory input from ipsilateral and contralateral limbs to F- and E-neurons located in different zones of grey matter differed from that observed in the control and after acute LHS (Fig. 11*E–H*).

Because there were differences in the strength of tilt-related sensory inputs from the ipsilateral and contralateral limb to F- and E-neurons in the control (reflected in the thickness of arrows in Fig. 11*E–H*; also, compare Fig. 8*B* and *C* and Fig. 9*B* and *C*, respectively), one can suggest that, on the ipsi-LHS side, the recovery of the response of F-neurons in zone 3 and E-neurons in zone 1 to whole platform tilts in R-rabbits (Fig. 11*B* and *D*) was caused largely by the recovery of input from the ipsilateral limb to F-neurons (red solid arrow to zone 3 in Fig. 11*F*) and by the recovery of input from the contralateral limb to E-neurons (red solid arrow to zone 1 in *Fig. 11H*). On the co-LHS side, a decrease in inputs from the ipsilateral limb to F-neurons in zones 1 and 2 (red dashed arrows to these zones in Fig. 11*F*) and the recovery of inputs from contralateral limb to E-neurons in zones 2 and 3 (red solid arrows to these zones in *Fig. 11H*) most likely contributed to the reduction and enhancement, respectively, of the responses to tilts observed in the local populations of these neurons in corresponding zones (Fig. 6*F* and *H*, respectively).

### Origin of correct and incorrect EMG responses to tilts in decerebrate R-rabbits

As demonstrated in the present study, in R-rabbits after decerebration, the EMG pattern of PLRs was similar to

control on the ipsi-LHS side but distorted on the co-LHS side. This was reflected in a substantial proportion of cycles in which EMG responses were absent or incorrectly phased (Fig. 2*C*). Weak residual inconsistent correct and incorrect EMG responses to tilts were observed after acute LHS on the ipsi-LHS side (Zelenin, Lyalka, Orlovsky et al. 2016) and after acute spinalization on both sides (Musienko et al. 2010; Zelenin et al. 2013; Zelenin, Lyalka, Hsu et al. 2016). To explain the origin of these responses, one of the possible hypothetical spinal circuitries has been proposed (Zelenin, Lyalka, Orlovsky et al. 2016) (Fig. 11*I* and *J*). The recovery and distortion of EMG patterns of PLRs observed on the ipsi-LHS and co-LHS sides, respectively, in R-rabbits can be explained in the framework of these hypothetical circuits (Fig. 11*K* and *L*).

One can suggest that the recovery of correct EMG responses to tilts on the ipsi-LHS side in R-rabbits is caused by plastic changes compensating for the effect of the acute LHS (Fig. 11*L*). They lead to an increase in the excitability level of interneurons F1, F2, E1 and E3, and motoneurons (MNs) (which was decreased after acute LHS; Fig. 11*J*), and the inhibition of interneurons F3 and E2 (for which the excitability level was increased after acute LHS) (Fig. 11*J*). As a result, platform tilts evoke correct EMG responses in extensors: they are activated during ipsilateral limb flexion (through the reflex chain E-Aff→F1→F2→MN) and inactivated during limb extension (through the reflex chain F-Aff→E1→E3→MN) (Fig. 11*L* and *N*). The changes in the excitability of the abovementioned neurons could be a result of the re-innervation of these neurons by supraspinal and/or propriospinal neurons with axons running on the co-LHS side of the spinal cord (Bareyre et al. 2004; Fenrich & Rose, 2009; Flynn et al. 2011; Oudega & Perez, 2012; Vavrek et al. 2006; Zörner et al. 2014). An increase in the level of excitability could also be a result of the development of denervation supersensitivity (Cannon, 1932; Thesleff & Sellin, 1980). Previously, we demonstrated that, after spinalization, the activity of F- and E-neurons in rabbits recovers in a few days, whereas the recovery of motoneuronal excitability takes months (Zelenin et al. 2019).

We demonstrated that, after the decerebration of R-rabbits, a substantial proportion of incorrect responses to tilts and a proportion of tilt cycles with an absence of EMG responses were observed on the co-LHS side (Fig. 2*C*). One can assume that, in R-rabbits, as a result of plastic changes, the level of excitability of brainstem supraspinal neurons, which, in the control, activate F2 and E3 and inhibit F3 and E2 (Fig. 11*I*), is strongly influenced by the input from the forebrain (Fig. 11*K*). Decerebration causes (i) an increase in the excitability of F3 and E2 neurons (as a result of a decrease in the inhibitory supraspinal drive); (ii) a decrease in the

excitability of F2 and E3 neurons (as a result of a decrease in the excitatory supraspinal drive); and (iii) no changes in the excitability of E1, F1 and MNs. Correct and incorrect motor responses to tilts observed after decerebration can be explained by the spontaneous fluctuations of F2, F3, E2 and E3 neuron excitability. If the excitability of F3 and E2 neurons is higher than that of F2 and E3 neurons, the flexion/extension movements of the limb will activate the reflex chains E-Aff→F1→F3→MN and F-Aff→E1→E2→MN, respectively, resulting in the generation of incorrect responses (Fig. 11*K* and *M*). If the excitability of F2 and E3 neurons is higher than that of F3 and E2 neurons (as in the control, Fig. 11*I*), the correct responses will be generated (Fig. 11*N*). If activities in F2 and F3 during limb flexion and activities in E2 and E3 during limb extension are equal, the summation of the equal excitatory and inhibitory inputs by MNs results in the absence of motor response to tilt. We demonstrated that a shift in the balance between the responses of F-population and E-population located in zones 2 and 3 from the predominance of F-population response toward the predominance of E-population response leads to an increase in the proportion of incorrect EMG responses and a decrease in the proportion of correct EMG responses to tilts after acute LHS, as well as in R-rabbits (Fig. 7). One can assume that the shift is caused by predominance in the activity/proportion in E2 over that in F2.

To conclude, in the present study, PLRs and the activity of spinal PLR-related neurons in animals with recovered after LHS postural functions have been characterized for the first time. A recovery of PLRs on the ipsi-LHS side and the substantial distortion of PLRs on the co-LHS side (almost unaffected by acute LHS) have been found after decerebration. We demonstrated that the activity of PLR-related neurons and the efficacy of posture-related sensory inputs to them mostly returned to the control level in areas of grey matter where they were significantly decreased after acute LHS. By contrast, these parameters were significantly changed compared to the control in areas where they were unaffected by acute LHS. We suggest that these changes underlay the recovery and distortion of PLRs on the ipsi-LHS and co-LHS side, respectively. The results obtained suggest that plastic changes underlying the recovery of postural functions after LHS take place in both postural networks affected and unaffected by acute LHS. Changes at the forebrain level substantially contribute to the generation of postural corrections on the co-LHS side, whereas plastic changes in the brainstem–cerebellum–spinal postural networks underlay the recovery of postural functions on the ipsi-LHS side. The results obtained advance our understanding of the neuronal mechanisms underlying the recovery of postural functions after LHS and can potentially be used for the development of novel strategies for the recuperation of postural functions in patients with spinal cord injury Supporting Infromation).

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

## Additional information

### Data availability statement

The data that support the findings of this study are available from the corresponding author upon reasonable request.

### Competing interests

The authors declare that they have no competing interests.

### Author contributions

T.G.D. and P.V.Z. designed the experiments. P.V.Z., V.F.L. and T.G.D. performed the experiments. V.F.L. and P.V.Z. analysed data. T.G.D. wrote the paper with contribution from all authors. All authors approved the final version of the manuscript submitted for publication.

## Funding

This work was supported by grants from NIH (R01 NS-064964) and the Swedish Research Council (2017-02944; 2020−02502) to TGD.

## Keywords

decerebrate rabbit, postural reflexes, spinal cord injury, spinal networks, spinal neurons

## Supporting information

Additional supporting information can be found online in the Supporting Information section at the end of the HTML view of the article. Supporting information files available:

**Statistical Summary Document**
**Peer Review History**
**Supporting information**

