## [Peer Review History · The Journal of Physiology]

Changes in operation of postural networks in rabbits with postural functions recovered after lateral hemisection of the spinal cord.

Pavel V Zelenin, Vladimir F. Lyalka, and Tatiana G Deliagina

DOI: 10.1113/JP283458

Corresponding author(s): Tatiana Deliagina (tatiana.deliagina@ki.se)

The following individual(s) involved in review of this submission have agreed to reveal their identity: T. Richard Nichols (Referee #1)

Review Timeline:

Submission Date:	19-Aug-2022
Editorial Decision:	22-Sep-2022
Revision Received:	18-Oct-2022
Editorial Decision:	22-Nov-2022
Revision Received:	28-Nov-2022
Accepted:	30-Nov-2022

Senior Editor: Richard Carson

Reviewing Editor: Jing-Ning Zhu

Transaction Report:

Dear Professor Deliagina,

Re: JP-RP-2022-283458 "Changes in operation of postural networks in rabbits with postural functions recovered after lateral hemisection of the spinal cord." by Pavel V Zelenin, Vladimir F. Lyalka, and Tatiana G Deliagina

Thank you for submitting your manuscript to The Journal of Physiology. It has been assessed by a Reviewing Editor and by 2 expert Referees and I am pleased to tell you that it is considered to be acceptable for publication following satisfactory revision.

The reports are copied at the end of this email. Please address all of the points and incorporate all requested revisions, or explain in your Response to Referees why a change has not been made.

NEW POLICY: In order to improve the transparency of its peer review process The Journal of Physiology publishes online as supporting information the peer review history of all articles accepted for publication. Readers will have access to decision letters, including all Editors' comments and referee reports, for each version of the manuscript and any author responses to peer review comments. Referees can decide whether or not they wish to be named on the peer review history document.

Authors are asked to use The Journal's premium BioRender (<https://biorender.com/>) account to create/redraw their Abstract Figures. Information on how to access The Journal's premium BioRender account is here:

<https://physoc.onlinelibrary.wiley.com/journal/14697793/biorender-access> and authors are expected to use this service. This will enable Authors to download high-resolution versions of their figures. The link provided should only be used for the purposes of this submission. Authors will be charged for figures created on this premium BioRender account if they are not related to this manuscript submission.

I hope you will find the comments helpful and have no difficulty returning your revisions within 4 weeks.

Your revised manuscript should be submitted online using the links in Author Tasks: Link Not Available.

Any image files uploaded with the previous version are retained on the system. Please ensure you replace or remove all files that have been revised.

REVISION CHECKLIST:

- Article file, including any tables and figure legends, must be in an editable format (eg Word)
- Abstract figure file (see above)
- Statistical Summary Document
- Upload each figure as a separate high quality file
- Upload a full Response to Referees, including a response to any Senior and Reviewing Editor Comments;
- Upload a copy of the manuscript with the changes highlighted.

- A potential 'Cover Art' file for consideration as the Issue's cover image;
- Appropriate Supporting Information (Video, audio or data set https://jp.msubmit.net/cgi-bin/main.plex?form_type=display_requirements#supp).

To create your 'Response to Referees' copy all the reports, including any comments from the Senior and Reviewing Editors, into a Word, or similar, file and respond to each point in colour or CAPITALS and upload this when you submit your revision.

I look forward to receiving your revised submission.

If you have any queries please reply to this email and staff will be happy to assist.

Yours sincerely,

Richard Carson
Senior Editor
The Journal of Physiology

REQUIRED ITEMS:

- Author photo and profile. First (or joint first) authors are asked to provide a short biography (no more than 100 words for one author or 150 words in total for joint first authors) and a portrait photograph. These should be uploaded and clearly labelled with the revised version of the manuscript. See Information for Authors for further details.
- You must start the Methods section with a paragraph headed Ethical Approval. A detailed explanation of journal policy and regulations on animal experimentation is given in Principles and standards for reporting animal experiments in The Journal of Physiology and Experimental Physiology by David Grundy *J Physiol*, 593: 2547-2549. doi:10.1113/JP270818). A checklist outlining these requirements and detailing the information that must be provided in the paper can be found at: <https://physoc.onlinelibrary.wiley.com/hub/animal-experiments>. Authors should confirm in their Methods section that their experiments were carried out according to the guidelines laid down by their institution's animal welfare committee, and conform to the principles and regulations as described in the Editorial by Grundy (2015). The Methods section must contain details of the anaesthetic regime: anaesthetic used, dose and route of administration and method of killing the experimental animals.
- Your manuscript must include a complete Additional Information section.
- The Journal of Physiology funds authors of provisionally accepted papers to use the premium BioRender site to create high resolution schematic figures. Follow this link and enter your details and the manuscript number to create and download figures. Upload these as the figure files for your revised submission. If you choose not to take up this offer we require figures to be of similar quality and resolution. If you are opting out of this service to authors, state this in the Comments section on the Detailed Information page of the submission form. The link provided should only be used for the purposes of this submission. Authors will be charged for figures created on this premium BioRender account if they are not related to this manuscript submission.
- Please upload separate high-quality figure files via the submission form.
- Please ensure that the Article File you upload is a Word file.
- A Statistical Summary Document, summarising the statistics presented in the manuscript, is required upon revision. It must be on the Journal's template, which can be downloaded from the link in the Statistical Summary Document section here: https://jp.msubmit.net/cgi-bin/main.plex?form_type=display_requirements#statistics.
- Papers must comply with the Statistics Policy: https://jp.msubmit.net/cgi-bin/main.plex?form_type=display_requirements#statistics.

In summary:

- If $n \leq 30$, all data points must be plotted in the figure in a way that reveals their range and distribution. A bar graph with data points overlaid, a box and whisker plot or a violin plot (preferably with data points included) are acceptable formats.
- If $n > 30$, then the entire raw dataset must be made available either as supporting information, or hosted on a not-for-profit repository e.g. FigShare, with access details provided in the manuscript.
- 'n' clearly defined (e.g. x cells from y slices in z animals) in the Methods. Authors should be mindful of pseudoreplication.
- All relevant 'n' values must be clearly stated in the main text, figures and tables, and the Statistical Summary Document (required upon revision).
- The most appropriate summary statistic (e.g. mean or median and standard deviation) must be used. Standard Error of the Mean (SEM) alone is not permitted.
- Exact p values must be stated. Authors must not use 'greater than' or 'less than'. Exact p values must be stated to three significant figures even when 'no statistical significance' is claimed.
- Statistics Summary Document completed appropriately upon revision.

- Please include an Abstract Figure. The Abstract Figure is a piece of artwork designed to give readers an immediate understanding of the research and should summarise the main conclusions. If possible, the image should be easily 'readable' from left to right or top to bottom. It should show the physiological relevance of the manuscript so readers can assess the importance and content of its findings. Abstract Figures should not merely recapitulate other figures in the manuscript. Please try to keep the diagram as simple as possible and without superfluous information that may distract from the main conclusion(s). Abstract Figures must be provided by authors no later than the revised manuscript stage and should be uploaded as a separate file during online submission labelled as File Type 'Abstract Figure'. Please ensure that you include the figure legend in the main article file. All Abstract Figures should be created using BioRender. Authors should use The Journal's premium BioRender account to export high-resolution images. Details on how to use and access the premium account are included as part of this email.

EDITOR COMMENTS

Reviewing Editor:

The manuscript demonstrates changes in operation of postural networks following acute lateral hemisection of the spinal cord, which contributes to the understanding of mechanism of postural control and plasticity of postural networks after spinal injury. Please extensively refine the sentences throughout the manuscript and provide an explicit statement of euthanasia according to the suggestions of Reviewer 1.

Senior Editor:

Please address all of the issues raised in the detailed reviews.

Although in many cases raw data have been provided, Referee #1 makes the point that more complete presentation of the EMG results is warranted.

REFEREE COMMENTS

Referee #1:

General Comments

This manuscript contains valuable data that follows a series of papers from this group on alterations of postural control after spinal cord injury. Postural responses as well as corresponding EMG and neuronal responses are described. It is shown that, following recovery from spinal hemisection, EMG responses were altered more on the untreated than the damaged side. Considerable detail on the responses of interneurons was included and forms the bulk of the analysis. Although this an important and groundbreaking study, it suffers from a number of weaknesses. These weaknesses can be readily addressed. The authors refer to the contralesional side as "unaffected," whereas the interconnections between sides of the spinal cord and the bilateral distribution of some descending systems indicate that both sides are affected. As in other cases detailed below, there could be a better choice of words, such as contralesional or untreated. There is not a full description of the EMG responses, and some that are illustrated in Figure 1 are not discussed. Possible mechanisms for increased interneuronal excitability should include some recent work from investigators at Northwestern University. The main weaknesses are editorial in nature. Some sentences are very long and confusing, a figure is mislabeled repeatedly in the text, and some data in the figures are not discussed in the text. In some cases the editorial issues lead to confusion, such as detailed in comments 1 and 8. The following specific comments are intended to help improve the clarity of the manuscript.

Specific Comments

1. (Abstract) Qualify the statement that the patterns of EMG were similar to control but yet the postural responses were altered. (see comment #8).
2. (p 5, paragraph 2) ...acute LHS hardly affects PLRs... (remove "almost does not affect").
3. (p 5, para 3) "postural corrections caused by lateral tilt... have been evaluated...".
4. (p 6, para 3) Did the animals receive any training or behavioral testing during the recovery period?
5. (p 8, para 1) Was the lever to detect lateral displacement of the caudal trunk spring loaded or attached to the trunk in any way? Either of these would seem necessary to detect motion in both directions.
6. (Methods). Were the animals perfused at the end of the acute experiment? I did not see a statement concerning

euthanasia. Such a statement should be provided.

7. Figure 1. The legend needs an explanation of the shaded regions in H and I, and arrows in I. Also, the muscle responses in the shaded regions for H and I are opposite despite the similar value of alpha. This part of the figure does not seem to be discussed in the text. It should be discussed in the text and in the figure legend.

8. (p 12, para 2). The data in Figures 1H and 1I need further explanation, with respect to the shaded areas and the arrows. See comment 6 above.

9. (p 12, para 2, Figure 2A). Description of the EMG responses could be clearer. The term "pattern" could refer to intermuscular distribution or wave shape. It would be helpful to distinguish these features from the direction of the response ("correct" or appropriate vs incorrect or inappropriate). It is stated that the EMG patterns in R-rabbits differed somewhat from those in controls, and yet the abstract states that the patterns of EMG were similar. These points need to be reconciled. Showing some specific examples would help. Were there any differences in the EMG responses other than "correct" or "incorrect" direction (wave shape or ratio of EMG magnitudes for the two muscles)? If the only differences were in the appropriateness of the direction of the response, perhaps be more specific and just refer to correctness, or appropriateness rather than the more general term "pattern." From Figure 1 it appears that the wave shapes were different, perhaps due to the differences in mechanical constraints. Some comments on these differences would be helpful.

10. (Figure 2) It would be helpful to label A and B as intact or similar designation, and C-E as Decerebrate.

11. (p 13, para 1). What is meant by "disintegrated?"

12. (p 13, para 1). Similarly, the word "distortion" is ambiguous. Why not simply say that responses were absent or inappropriate in such and such cases. "Distortion" implies some change in the waveshape of the EMG.

13. (p 16, para 1, line 13) t[h]ree

14. (p 17, para 2) In this summary paragraph, it would help the reader to summarize the acute changes as well as the alterations occurring with recovery.

15. (p 21, section title) Changes in postural networks below forebrain unde[r]lay... or underlie?

16. (p 22, para 2). On the subject of recovery of responses and increased excitability of neurons, the work of David Bennett, Vicky Tysseling, and coworkers should be cited and discussed. These authors demonstrated increased firing in subsets of interneurons in the deep dorsal horn following spinal section and hemisection that were correlated with increased excitability of motoneurons, as well as evidence that these changes were related to alterations in serotonin input from the brainstem and changes in 5HT receptors. The data shown in Figure 6 indicate areas of increased activity in the deep dorsal horn that may result from the mechanisms revealed by Tysseling et al. It may be that the increased activity on the two sides results from different mechanisms, increases in descending input on the untreated side, as suggested here, and by increased excitability due to the reduction in descending serotonin on the treated side.

Bursting interneurons in the deep dorsal horn develop increased excitability and sensitivity to serotonin after chronic spinal injury.

Thaweerattanasin P, Birch D, Jiang MC, Tresch MC, Bennett DJ, Heckman CJ, Tysseling VM. *J Neurophysiol.* 2020 May 1;123(5):1657-1670. doi: 10.1152/jn.00701.2019. Epub 2020 Mar 25

17. (p 23, para 1). Again, I would be more specific about the changes in PLR's on the intact side in R-rabbits by simply stating their absence or inappropriateness and avoiding the term "pattern."

18. (p 23, para 2). "Considering that" rather than "considered that."

19. (p 25, section title). There seems to be a phrase missing in this section title. Also, the untreated side of the spinal cord is probably not unaffected, due to the exchange of information between the two sides and the presumed reduction in input from the damaged side.

20. (p 27, para 2). Is an example of an "incorrect" response shown in Figure 1 I? This shows the opposite response to those in 1H.

21. (p 27). The figure reference seems to be wrong. I assume you were referring to Figure 11 when citing Figure 9.

Referee #2 (please see attachment):

The impact of the present data in itself may be considered to be modest, but within the context of previous publications and providing a roadmap for further studies enhances its potential impact.

The insight provided in the discussion of other relevant work is extremely helpful in understanding the importance of this particular combination of data relative to previous publications.

The originality of the present research is without question.

The study design and the robustness of the experimental data reflect a thorough understanding and careful control of the important variables needed in the area of postural control.

The data presented and the manner that it is interpreted are consistent with the conclusions.

END OF COMMENTS

Confidential Review

19-Aug-2022

This manuscript provides data, interpretation and insight on mechanisms related to the anatomical and functional (postural responses), adaptations that occur largely in the spinal cord of adult cats following hemisection (LHS) at T 12. Similar functional and anatomical measures have been published in the uninjured, adult cat and also in the acute stage immediately following the same surgical procedure, demonstrated in how these properties change after an immediate loss of control derived from the lateral half of the spinal cord at the same vertebral level, T12. The present manuscript describes a subsequent experiment in which the same procedures were used, but the assessment of the adaptations described are those which occurred spontaneously 49-93 days after LHS, following the hemisection. Comparisons of these three experimental conditions (uninjured, acutely injured, and chronically injured) were compared with the performance of the three conditions been performed as three different experiments, but with most of the experimental procedures and outcomes being performed rather identically. It is reasonable to assume that these comparisons can be made. And for the most part, it would be very difficult to perform all of these procedures under different conditions at the same time. Confidence of this strategy seems justified given the unique, an extensive experience were performed largely by the same individuals with a level of expertise that is unlikely to occur in any other laboratory. Upon the completion of these three experimental protocols, the authors point out several key observations.

1. That the adaptations that occur post surgery are characterized as being dramatically distorted take place in postural networks that are injured at the acute stage, and the adaptive responses of the postural networks on the contralateral side of the lesion, i.e., a uninjured control state, were relatively few.
2. They demonstrate that changes that took place during the **recovery** period of postural functions take place not only in postural networks that are severely impaired, but also in those, that are almost unaffected by acute LHS.
3. Based on data obtained after decerebration, postural, responses on the damaged side recover mainly due to changes at the brainstem-cerebellum-spinal levels, suggesting that the forebrain is substantially involved in generation of the postural responses that occurred on the intact side during spontaneous recovery. In the decerebrated, recovered rabbits the postural responses were substantially distorted on the intact side while they were similar to control after acute LHS. Although this observation is perhaps expected, it has not been specifically demonstrated as clearly as the present set of data. This observation is important because it demonstrates another example of the complexity of the plasticity that occurs at multiple sites following an injury, which should be carefully considered in assuming what might be considered an appropriate control as representing an uninjured state.
4. Another significant observation was that in the acute stage was that while there was a significant decrease in the activity of spinal neurons in the gray matter on the injured side at the acute stage, it recovered to control levels, while in areas unaffected by acute LHS, was significantly changed. This reflects another important point regarding interpretation of data at a specific time point during which it is highly likely that the adaptive events continue to occur over prolonged periods, even when the behavioral outcome may be similar.

The authors carefully assess the present data with respect to the possible mechanisms that explain these adaptive events, but carefully conclude that there are multiple mechanisms that could have attributed to the combination of observations made. Based on that summation a logical conclusion is that there are multiple adaptive mechanisms in play over a prolonged period after a severe trauma to the spinal cord. The final sentence of the discussion could be viewed as being rather insignificant compared to the significance of the body of information presented.

Response to comments

Reviewing Editor

Comment 1. (i) Please extensively refine the sentences throughout the manuscript and (ii) provide an explicit statement of euthanasia according to the suggestions of Reviewer 1.

Response 1. (i) The manuscript was checked and edited by a proofreading service. (ii) The statement of euthanasia is added ((P11, Para4, L2-4).

Senior Editor:

Comment 1. Please address all of the issues raised in the detailed reviews.

Response 1. In the revised manuscript, all the issues raised by Reviewers are addressed.

Comment 2. Although in many cases raw data have been provided, Referee #1 makes the point that more complete presentation of the EMG results is warranted.

Response 2. The description of EMG results has been detailed and clarified (P12, Para3, L1-2; P12, Para4, L5-6; P13, Para2-3).

Reviewer 1

General comments

This manuscript contains valuable data that follows a series of papers from this group on alterations of postural control after spinal cord injury. Postural responses as well as corresponding EMG and neuronal responses are described. It is shown that, following recovery from spinal hemisection, EMG responses were altered more on the untreated than the damaged side. Considerable detail on the responses of interneurons was included and forms the bulk of the analysis. Although this an important and groundbreaking study, it suffers from a number of weaknesses. These weaknesses can be readily addressed. (i) The authors refer to the contralesional side as "unaffected," whereas the interconnections between sides of the spinal cord and the bilateral distribution of some descending systems indicate that both sides are affected. As in other cases detailed below, there could be a better choice of words, such as contralesional or untreated. (ii) There is not a full description of the EMG responses, and some that are illustrated in Figure 1 are not discussed. (iii) Possible mechanisms for increased interneuronal excitability should include some recent work from investigators at Northwestern University. (iv) The main weaknesses are editorial in nature. Some sentences are very long and confusing, (v) a figure is mislabeled repeatedly in the text, and (vi) some data in the figures are not discussed in the text. (vii) In some cases the editorial issues lead to confusion, such as detailed in comments 1 and 8. The following specific comments are intended to help improve the clarity of the manuscript.

Responses.

(i) We do not use the term "unaffected" to indicate the side which is contralateral to the lesion. We use the term "unaffected" or "almost unaffected" in relation to PLRs or to specific areas of the gray matter where activity of spinal neurons does not change after acute LHS. We agree with Reviewer that terms "damaged side" and "intact side" (which we used to indicate the side which is ipsilateral to the lesion, and the side which is contralateral to the lesion, respectively) may create confusion, since LHS affects activity of neurons in some areas of the

gray matter on the “intact” side. To avoid confusion, we replaced terms “intact” and “damaged” side by the “co-LHS” and “ipsi-LHS” side throughout the text.

- (ii) In the revised manuscript, the description of EMG responses has been revised and a detailed description of Fig.1 is provided (P.13, Para2,3).
- (iii) Suggested by Reviewer reference are added to the revised manuscript (P23, Para2, L11-15). See also the response to *Specific comment 16*.
- (iv) The manuscript was checked and edited by a proofreading service.
- (v) We apologize for misprints in Figure 11 references in Discussion. The misprints were corrected in the revised manuscript (P27, Para3, L3, Para 4, L5,6; P28, Para 1, L2,3,4; P 28, Para 2 L8,10; P 28, Para3, L3,4,5,8; P29, Para2, L5,6,14,16).
- (vi) In the revised manuscript a detailed description of Figs. is provided and unnecessary designations are removed (see responses to specific comments).
- (vii) See responses to Specific comments 1 and 8.

Specific comments

Comment 1. (Abstract) Qualify the statement that the patterns of EMG were similar to control but yet the postural responses were altered. (see comment #8).

Response 1. In Abstract, the statement related to EMG responses to postural disturbances in R-rabbits observed before decerebration is corrected (P3, L8-10).

Comment 2. (p 5, paragraph 2) ...acute LHS hardly affects PLRs... (remove "almost does not affect").

Response 2. Changed as suggested by Reviewer (P5, Para2, L7).

Comment 3. (p 5, para 3) "postural corrections caused by lateral tilt... have been evaluated...".

Response 3. The sentence was edited (P5, Para3, L4).

Comment 4. (p 6, para 3) Did the animals receive any training or behavioral testing during the recovery period?

Response 4. The animals were not subjected to training or any specific behavioral testing during recovery period. The postural test was performed just before the animal was taken to an acute experiment. This is clarified in the revised manuscript (P6, Para 3, L1-2).

Comment 5. (p 8, para 1) Was the lever to detect lateral displacement of the caudal trunk spring loaded or attached to the trunk in any way? Either of these would seem necessary to detect motion in both directions.

Response 5. To detect motion of the body in relation to the tilting platform in both directions a lever was kept pressed to the body with a soft spring. This is clarified in the revised manuscript (P8, Para1, L10-12).

Comment 6. (Methods). Were the animals perfused at the end of the acute experiment? I did not see a statement concerning euthanasia. Such a statement should be provided.

Response 6. The animals were perfused at the end of the acute experiments that caused euthanasia. This information is added in the revised manuscript (P11, Para4, L2-4).

Comment 7. (i) Figure 1. The legend needs an explanation of the shaded regions in H and I, and arrows in I. (ii) Also, the muscle responses in the shaded regions for H and I are opposite despite the similar value of alpha. This part of the figure does not seem to be discussed in the text. It should be discussed in the text and in the figure legend.

Response 7. (i) In Figure 1, shaded column in *H* and *I* as well as arrows in *I* are removed, as unnecessary.

(ii) The phases of the correct EMG response in the tilt cycle in the freely standing rabbit and in the decerebrate rabbit with a fixed spine are opposite. This is clarified in Results (P13, Para2) and Fig. 1 legend (P37, L19-25) in the revised manuscript.

Comment 8. (p 12, para 2). The data in Figures 1H and 1I need further explanation, with respect to the shaded areas and the arrows. See comment 6 above.

Response 8. In Figure 1, shaded columns in *H* and *I* as well as arrows in *I* are removed, as unnecessary.

Comment 9. (i) (p 12, para 2, Figure 2A). Description of the EMG responses could be clearer. The term "pattern" could refer to intermuscular distribution or wave shape. It would be helpful to distinguish these features from the direction of the response ("correct" or appropriate vs incorrect or inappropriate). (ii) It is stated that the EMG patterns in R-rabbits differed somewhat from those in controls, and yet the abstract states that the patterns of EMG were similar. These points need to be reconciled. Showing some specific examples would help. (iii) Were there any differences in the EMG responses other than "correct" or "incorrect" direction (wave shape or ratio of EMG magnitudes for the two muscles)? If the only differences were in the appropriateness of the direction of the response, perhaps be more specific and just refer to correctness, or appropriateness rather than the more general term "pattern." From Figure 1 it appears that the wave shapes were different, perhaps due to the differences in mechanical constraints. Some comments on these differences would be helpful.

Response 9. (i) We agree that the term "pattern" could be confusing since it could mean not only the phase of the EMG response in the tilt cycle which we analyzed, but also intermuscular distribution or wave shape. In the revised manuscript we replaced the term "EMG pattern" by "the phase of the EMG response in the tilt cycle" (P3, L8-10; P12, Para3, L1-2; P12, Para4, L5-6) or clarified that the term "pattern" is used in terms of the phases of EMG responses in the tilt cycle (P22, Para 3, L1-2).

(ii) In Abstract, the statement related to EMG responses to postural disturbances in R-rabbits observed before decerebration is corrected (P3, L8-10). The slight difference in EMG responses to tilts is illustrated in Fig. 2A.

(iii) We do not quite understand which comparison Reviewer means: a comparison of wave shapes or ratio of EMG magnitudes for the two muscles in control animals and R-rabbits or comparison of EMGs on the left and right side in R-rabbits or comparison of EMG responses before and after decerebration in R-rabbits. Unfortunately, with time, there is a deterioration of recording quality by chronically implanted EMG electrodes. This deterioration could be different for different muscles that prevents analysis of changes in ratio of EMG magnitudes in different muscles as well as changes in wave shapes of responses in individual muscles as compared to control as well as corresponding analysis for EMGs on the ipsi-LHS and co-LHS sides in R-rabbits. Also, comparison of EMG responses in freely standing R-rabbits and in fixed decerebrate R-rabbits is problematic due to substantial differences in mechanical

constrains. That is why we compared only the most robust parameter of the response that is the phases of EMG responses in the tilt cycle.

Comment 10. (Figure 2) It would be helpful to label A and B as intact or similar designation, and C-E as Decerebrate.

Response 10. According to suggestion by Reviewer, we labeled Fig. 2A,B “Before decerebration” and Fig. 2 C-E “After decerebration”.

Comment 11. (p 13, para 1). What is meant by "disintegrated?"

Response 11. The word “disintegrated” is removed. The corresponding part of the text is rewritten (P13, Para2,3).

Comment 12. (p 13, para 1). Similarly, the word "distortion" is ambiguous. Why not simply say that responses were absent or inappropriate in such and such cases. "Distortion" implies some change in the waveshape of the EMG.

Response 12. The word “distortion” is removed. As suggested by Reviewer, the corresponding sentence is reformulated (P13, Para3, L9-14).

Comment 13. (p 16, para 1, line 13) t[h]ree

Response 12. The misprint is corrected (P16, Para4, last line).

Comment 14. (p 17, para 2) In this summary paragraph, it would help the reader to summarize the acute changes as well as the alterations occurring with recovery.

Response 14. In the main body of this section, a detailed description of changes in specific areas of the grey matter observed during acute LHS as well as subsequent changes in affected and unaffected by acute LHS areas observed in R-rabbits are presented. Since in R-rabbits changes in activity of F- and E- neurons in areas affected as well as unaffected by acute LHS are different, it is impossible to summarize them briefly. We do not think that a repeated description of all changes will help the reader. The last paragraph contains a general conclusion of the observed changes. We prefer to avoid repetition of specific changes and to keep the text as it is.

Comment 15. (p 21, section title) Changes in postural networks below forebrain unde[r]lay... or underlie?

Response 15. The misprint is corrected. It was meant “underlie” (P22, section title).

Comment 16. (p 22, para 2). (i) On the subject of recovery of responses and increased excitability of neurons, the work of David Bennett, Vicky Tysseling, and coworkers should be cited and discussed. These authors demonstrated increased firing in subsets of interneurons in the deep dorsal horn following spinal section and hemisection that were correlated with increased excitability of motoneurons, as well as evidence that these changes were related to alterations in serotonin input from the brainstem and changes in 5HT receptors. (ii) The data shown in Figure 6 indicate areas of increased activity in the deep dorsal horn that may result from the mechanisms revealed by Tysseling et al. It may be that the increased activity on the two sides results from different mechanisms, increases in descending input on the untreated side, as suggested here, and by increased excitability due to the reduction in descending serotonin on the treated side.

Bursting interneurons in the deep dorsal horn develop increased excitability and sensitivity to serotonin after chronic spinal injury.

Thaweerattanasin P T, Birch D, Jiang MC, Tresch MC, Bennett DJ, Heckman CJ, Tysseling VM. *J Neurophysiol.* 2020 May 1;123(5):1657-1670. doi: 10.1152/jn.00701.2019. Epub 2020 Mar 25

Response 16. (i) We are grateful to Reviewer for the reminder of Tysseling's and Bennett's study. It is included in Discussion of the revised manuscript (P23, Para2, L11-15).

(ii) We don't quite agree with Reviewer, that "...the increased activity on the two sides results from different mechanisms, increases in descending input on the untreated side, as suggested here, and by increased excitability due to the reduction in descending serotonin on the treated side". First, in R-rabbits, a significant increase in activity of E-neurons in the area that coincides with location of the deep dorsal horn interneurons (described by Tysseling's and Bennett's groups) was observed only on the intact (untreated) side of the spinal cord (Fig. 6D). Thus, it can hardly be caused by "increased excitability due to the reduction in descending serotonin on the treated side". Second, re-innervation of spinal neurons located below the lesion by supraspinal and propriospinal neurons with axons running on the intact side of the spinal cord is well documented (Bareyre *et al.* 2004; Vavrek *et al.* 2006; Fenrich & Rose, 2009; Flynn *et al.* 2011; Oudega & Perez, 2012; Zörner *et al.* 2014). Thus, as indicated in the manuscript (P22, Para2), both mechanisms, an increase in the contralateral descending input and increased excitability due to loss of the ipsilateral descending input, most likely contribute to the increase in the activity of neurons on the "treated" side.

Comment 17. (p 23, para 1). Again, I would be more specific about the changes in PLR's on the intact side in R-rabbits by simply stating their absence or inappropriateness and avoiding the term "pattern."

Response 17. Changed as suggested by Reviewer (P24, Para2. L1-3).

Comment 18. (p 23, para 2). "Considering that" rather than "considered that."

Response 18. Changed as suggested by Reviewer (P24, Para2, L1).

Comment 19. (p 25, section title). There seems to be a phrase missing in this section title. Also, the untreated side of the spinal cord is probably not unaffected, due to the exchange of information between the two sides and the presumed reduction in input from the damaged side.

Response 19. The section title is corrected (P26). We agree with Reviewer, that LHS affects activity of neurons located on both sides of the spinal cord. However, we demonstrated that the affected neurons are located in specific areas of the grey matter. On both sides, there are areas where neurons exhibit a significant change in the activity after acute LHS (areas affected by LHS), as well as areas in which the activity of neurons is similar to control (areas unaffected by LHS) (Fig. 6A,B).

Comment 20. (p 27, para 2). Is an example of an "incorrect" response shown in Figure 1 I? This shows the opposite response to those in 1H.

Response 20. The phases of a "correct" extensor muscle response in the tilt cycle in the intact freely standing rabbit and in the decerebrate rabbit with the fixed spine are opposite (see *Response 8*). Thus, all EMG responses in Fig. 1H and EMG responses in Vast-L, Vast-R, and Gast-L in Fig. 1I are "correct" responses. On P27, Para2, L1-4, we refer to Fig. 2C, where a proportion of cycles in which EMG responses were absent or incorrectly phased are shown.

Comment 21. (p 27). The figure reference seems to be wrong. I assume you were referring to

Figure 11 when citing Figure 9.

Response 21. We apologize for misprints in Figure 11 reference in Discussion. The misprints were corrected in the revised manuscript (P27, Para3, L3, Para 4, L5,6; P28, Para 1, L2,3,4; P 28, Para 2 L8,10; P 28, Para3, L3,4,5,8; P29, Para2, L5,6,14,16).

Reviewer 2

Comment 1. The final sentence of the discussion could be viewed as being rather insignificant compared to the significance of the body of information presented.

Response 1. We are thankful to Reviewer 2 for appreciation of our study. We agree with Reviewer that the last sentence in Discussion is not important for experts in the area. However, we prefer to keep it to emphasize the significance of our study for a wider audience working in different areas of physiology or whose research crosses boundaries between physiology and other fields.

Dear Professor Deliagina,

Re: JP-RP-2022-283458R1 "Changes in operation of postural networks in rabbits with postural functions recovered after lateral hemisection of the spinal cord." by Pavel V Zelenin, Vladimir F. Lyalka, and Tatiana G Deliagina

Thank you for submitting your manuscript to The Journal of Physiology. It has been assessed by a Reviewing Editor and by 1 expert referee and we are pleased to tell you that it is acceptable for publication following minor revision.

REVISION CHECKLIST:

- 'Potential Cover Art' for consideration as the issue's cover image

- Appropriate Supporting Information (Video, audio or data set: see https://jp.msubmit.net/cgi-bin/main.plex?form_type=display_requirements#supp).

We look forward to receiving your revised submission.

Yours sincerely,

Richard Carson
Senior Editor
The Journal of Physiology

REQUIRED ITEMS:

- The Methods section must contain details of the anaesthetic regime: anaesthetic used, dose and route of administration and method of killing the experimental animals.
- Your manuscript must include a complete Additional Information section.
- The Journal of Physiology funds authors of provisionally accepted papers to use the premium BioRender site to create high resolution schematic figures. Follow this link and enter your details and the manuscript number to create and download figures. Upload these as the figure files for your revised submission. If you choose not to take up this offer we require figures to be of similar quality and resolution. If you are opting out of this service to authors, state this in the Comments section on the Detailed Information page of the submission form. The link provided should only be used for the purposes of this submission. Authors will be charged for figures created on this premium BioRender account if they are not related to this manuscript submission.
- Papers must comply with the Statistics Policy: https://jp.msubmit.net/cgi-bin/main.plex?form_type=display_requirements#statistics.

In summary:

- If $n \leq 30$, all data points must be plotted in the figure in a way that reveals their range and distribution. A bar graph with data points overlaid, a box and whisker plot or a violin plot (preferably with data points included) are acceptable formats.
- If $n > 30$, then the entire raw dataset must be made available either as supporting information, or hosted on a not-for-profit repository e.g. FigShare, with access details provided in the manuscript.
- 'n' clearly defined (e.g. x cells from y slices in z animals) in the Methods. Authors should be mindful of pseudoreplication.
- All relevant 'n' values must be clearly stated in the main text, figures and tables, and the Statistical Summary Document (required upon revision).
- The most appropriate summary statistic (e.g. mean or median and standard deviation) must be used. Standard Error of the Mean (SEM) alone is not permitted.
- Exact p values must be stated. Authors must not use 'greater than' or 'less than'. Exact p values must be stated to three significant figures even when 'no statistical significance' is claimed.
- Statistics Summary Document completed appropriately upon revision.

EDITOR COMMENTS

Reviewing Editor:

Thanks for the efforts of the authors to address all the issues raised by the reviewer.

Senior Editor:

Upon inspection of the information provided in the statistical summary document, it appears that some analyses have been conducted on the basis of the individual responses (i.e., with sample size n), rather than using a multi-level design, whereby

the individual samples are nested within subjects (N). The use of n samples (rather than n nested within N) will dramatically increase the degrees of freedom.

Please ensure that all analyses are conducted again using the appropriate multi-level design. It is recognised that this is likely to require entirely different statistical procedures. Dependent on the pattern of outcomes that emerges, the modification of not only the Results section, but also the Discussion may be indicated.

REFEREE COMMENTS

Referee #1:

The authors have satisfactorily addressed all of the comments from this reviewer.

END OF COMMENTS

1st Confidential Review

18-Oct-2022

Response to comments

Senior Editor:

Upon inspection of the information provided in the statistical summary document, it appears that some analyses have been conducted on the basis of the individual responses (i.e., with sample size n), rather than using a multi-level design, whereby the individual samples are nested within subjects (N). The use of n samples (rather than n nested within N) will dramatically increase the degrees of freedom.

Please ensure that all analyses are conducted again using the appropriate multi-level design. It is recognised that this is likely to require entirely different statistical procedures. Dependent on the pattern of outcomes that emerges, the modification of not only the Results section, but also the Discussion may be indicated.

Response:

We agree that pooling of data obtained in different animals (when we characterized percentage of different EMG responses) inflates the degrees of freedom. We reanalyzed our data related to Fig. 2 and confirmed that all our conclusions are valid despite the inter-subject variability. The corresponding changes are made in the statistical summary document and in the manuscript text.

As for the characteristics of the neuronal activity, they cannot be subjected to “multi-level design” statistical analysis because of the nature of the experimental data. It was technically impossible to systematically record neurons across the entire gray matter in each individual animal. Thus, the data samples obtained in an individual animal could not be representative for the entire neuronal population across the transversal section of the gray matter. This fact makes any inter-subject comparisons inappropriate. We explicitly mention this in Methods: “Unfortunately, it was impossible to explore systematically the entire gray matter in individual decerebrate rabbits. That is why we pooled all neurons recorded on the damaged side and all neurons recorded on the intact side from 8 rabbits and did not perform any cross-subject comparison.” (P9, Para3, L6-9). Other authors who record activity of individual neurons encounter the same problem of insufficient quantity of data. These authors need to pool data from many animals and do not analyze the inter-subject variability. One can find this in many articles recently published in the Journal of Physiology, e.g.:

J Physiol. 2022 Jun;600(12):2939-2952. doi: 10.1113/JP282873

J Physiol. 2022 Jul;600(13):3149-3167. doi: 10.1113/JP282509

J Physiol. 2022 Jul;600(14):3355-3381. doi: 10.1113/JP282804

Dear Dr Deliagina,

Re: JP-RP-2022-283458R2 "Changes in operation of postural networks in rabbits with postural functions recovered after lateral hemisection of the spinal cord." by Pavel V Zelenin, Vladimir F. Lyalka, and Tatiana G Deliagina

We are pleased to tell you that your paper has been accepted for publication in The Journal of Physiology.

Authors should note that it is too late at this point to offer corrections prior to proofing. The accepted version will be published online, ahead of the copy edited and typeset version being made available. Major corrections at proof stage, such as changes to figures, will be referred to the Editors for approval before they can be incorporated. Only minor changes, such as to style and consistency, should be made at proof stage. Changes that need to be made after proof stage will usually require a formal correction notice.

Yours sincerely,

Richard Carson
Senior Editor
The Journal of Physiology

P.S. - You can help your research get the attention it deserves! Check out Wiley's free Promotion Guide for best-practice recommendations for promoting your work at www.wileyauthors.com/eeo/guide. You can learn more about Wiley Editing Services which offers professional video, design, and writing services to create shareable video abstracts, infographics, conference posters, lay summaries, and research news stories for your research at www.wileyauthors.com/eeo/promotion.

IMPORTANT NOTICE ABOUT OPEN ACCESS: To assist authors whose funding agencies mandate public access to published research findings sooner than 12 months after publication, The Journal of Physiology allows authors to pay an Open Access (OA) fee to have their papers made freely available immediately on publication.

You can check if your funder or institution has a Wiley Open Access Account here: <https://authorservices.wiley.com/author-resources/Journal-Authors/licensing-and-open-access/open-access/author-compliance-tool.html>.

EDITOR COMMENTS

Reviewing Editor:

I have no further comments.